# Enhancing and shaping the immunogenicity of native-like HIV-1 envelope trimers with a two-component protein nanoparticle

Philip J.M. Brouwer [1], Aleksandar Antanasijevic[2], Zachary Berndsen[2], Anila Yasmeen[3], Brooke Fiala[4,5], Tom P.L. Bijl[1], Ilja Bontjer[1], Jacob B. Bale [4,5,10], William Sheffler[4,5], Joel D. Allen [6], Anna Schorcht[1], Judith A. Burger[1], Miguel Camacho[1], Daniel Ellis[4,5], Christopher A. Cottrell [2], Anna-Janina Behrens[7,11], Marco Catalano[1], Iván del Moral-Sánchez[1], Thomas J. Ketas[3], Celia LaBranche[8], Marit J. van Gils [1], Kwinten Sliepen [1], Lance J. Stewart[4,5], Max Crispin[6,7], David C. Montefiori[8], David Baker [4,5,9], John P. Moore [3], Per Johan Klasse[3], Andrew B. Ward [2], Neil P. King[4,5] & Rogier W. Sanders[1]

The development of native-like HIV-1 envelope (Env) trimer antigens has enabled the induction of neutralizing antibody (NAb) responses against neutralization-resistant HIV-1 strains in animal models. However, NAb responses are relatively weak and narrow in specificity. Displaying antigens in a multivalent fashion on nanoparticles (NPs) is an established strategy to increase their immunogenicity. Here we present the design and characterization of two-component protein NPs displaying 20 stabilized SOSIP trimers from various HIV-1 strains. The two-component nature permits the incorporation of exclusively well-folded, native-like Env trimers into NPs that self-assemble in vitro with high efficiency. Immunization studies show that the NPs are particularly efficacious as priming immunogens, improve the quality of the Ab response over a conventional one-component nanoparticle system, and are most effective when SOSIP trimers with an apex-proximate neutralizing epitope are displayed. Their ability to enhance and shape the immunogenicity of SOSIP trimers make these NPs a promising immunogen platform.

[1] Amsterdam UMC, Department of Medical Microbiology, Amsterdam Infection & Immunity Institute, University of Amsterdam, Amsterdam 1105AZ, The Netherlands. [2] Department of Integrative Structural and Computational Biology, The Scripps Research Institute, La Jolla, California 92037, USA. [3] Department of Microbiology and Immunology, Weill Medical College of Cornell University, New York, New York 10065, USA. [4] Department of Biochemistry, University of Washington, Seattle, Washington 98195, USA. [5] Institute for Protein Design, University of Washington, Seattle, Washington 98195, USA. [6] Biological Sciences and Institute of Life Sciences, University of Southampton, SO17 1BJ Southampton, UK. [7] Department of Biochemistry, Oxford Glycobiology Institute, University of Oxford, OX1 3QU Oxford, UK. [8] Department of Surgery, Duke University Medical Center, Durham, North Carolina 27710, USA. [9] Howard Hughes Medical Institute, University of Washington, Seattle, Washington 98105, USA. [10] Present address: Arzeda Corporation, Seattle, Washington 98119, USA. [11] Present address: New England Biolabs, Inc., Ipswich, Massachussetts 01938, USA. Correspondence and requests for materials should be addressed to N.P.K. (email: neilking@uw.edu) or to R.W.S. (email: r.w.sanders@amc.uva.nl)

To counter the extensive sequence diversity of HIV-1, it is widely accepted that a successful vaccine will need to induce broadly neutralizing antibodies (bNAbs)[1–3]. Immunogens intended to induce bNAbs are all based on the viral envelope glycoproteins (Env) and an increasingly commonly used design platform involves SOSIP trimers, which are recombinant, soluble mimics of Env[3–5]. However, despite their ability to elicit autologous NAbs against resistant (Tier-2) HIV-1 strains in animal models, further improvements to the design and delivery of SOSIP trimers are required to increase the potency, longevity, and breadth of NAb responses[6–8].

Displaying antigens in a particulate array on synthetic nanoparticles (NPs) or, alternatively, virus-like particles (VLPs) is an established strategy to increase their immunogenicity[9–12]. Multivalent antigens that can mimic the repetitive and well-ordered antigenic structures found on many pathogens cross-link B-cell receptors (BCRs) and activate B cells more efficiently than their monovalent counterparts[10–12]. In addition, they can be taken up by antigen-presenting cells and trafficked to lymph nodes more efficiently, leading to improved formation of germinal centers[13–16]. This category of immunogen may be particularly valuable for priming responses, because the avidity advantage conferred by multivalent presentation may compensate for interactions with the BCRs of naive B cells, which tend to be of low affinity.

Self-assembling proteins are attractive as scaffolds for multivalent antigen presentation, as they form well-defined materials that seamlessly integrate the displayed antigen via genetic fusion, enabling large-scale production by standard methods for producing biologics[9]. Env trimers have previously been presented in mutlivalent form on ferritin NPs[17–20]. However, these NPs assemble intracellularly; a process that can lead to the presence of misfolded and non-native trimers on the secreted particles, and hence the presentation of unwanted epitopes for non-neutralizing antibodies (non-NAbs)[17,21,22]. In addition, furin-mediated cleavage of Env is inefficient on ferritin particles[17,18]. Perhaps in consequence, Env-ferritin particles induced NAb responses that were skewed towards neutralization-sensitive Tier-1 viruses[17,18,22]. Env trimers have also been expressed on enveloped VLPs. However, the same issue surrounding intracellular assembly applies here as well, and HIV-1 Env VLPs are difficult to produce in large amounts and to an appropriate degree of homogeneity and purity[23–25].

Alternatively, purified, recombinant soluble Env proteins, including native-like trimers, can be non-covalently or covalently attached to liposomes or, as recently demonstrated, covalently attached to ready-formed VLPs to make particulate antigens[15,16,26–29]. The practicalities of producing liposome-based immunogens on a large scale for human trials remain to be addressed.

We recently reported the computational design of self-assembling protein NPs constructed from two distinct oligomeric protein components[30]. This property makes it possible to express and purify each component separately; when mixed, they assemble efficiently into highly ordered, homogenous icosahedral particles ranging from 24 to 40 nm in diameter. The enhanced control offered by designed two-component NPs could be useful for the production of NP immunogens displaying exclusively well-folded, native-like HIV-1 Env trimers at high density. Such immunogens would circumvent the conformational heterogeneity and low density of Env on the viral membrane, both of which are proposed immune evasion mechanisms of HIV-1[31,32]. Here we describe the computational design, production, and in vitro properties of self-assembling, two-component NP immunogens that present native-like SOSIP trimers of several genotypes. We also report on the immunogenicity in rabbits of NP immunogens that present BG505 SOSIP.v5.2 or ConM SOSIP.v7 trimer.

## Results

### BG505 SOSIP-I53-50A.1NT1 comprises native-like trimers.

We recently described the computational design of two-component self-assembling nanomaterials that can be assembled in vitro from independently purified subunits[30,33]. To generate building blocks capable of displaying native-like Env trimers, we computationally docked the crystal structure of BG505 SOSIP.664[34] to a set of trimeric NP components with exterior-facing N termini (see Methods). An early screen for the expression of several of our designed nanoparticle components fused to BG505 SOSIP revealed that the trimeric component of the recently described NP I53-50[30], designated I53-50A, secreted at higher levels than any other scaffold when fused to BG505 SOSIP. Selecting this scaffold for experimental evaluation was further supported by the observation that I53-50A is extremely stable and capable of scaffolding viral glycoprotein antigens[35]. I53-50NP is a 120-subunit assembly with icosahedral symmetry comprising 20 trimeric (I53-50A) and 12 pentameric (I53-50B) building blocks; a NP immunogen constructed from a SOSIP-I53-50A trimeric fusion protein (Fig. 1a) would present 20 SOSIP trimers in a dense array on the NP exterior.

We genetically fused the previously described BG505 SOSIP.v5.2 sequence[36] to the prototypical I53-50A and three variants of this component: I53-50A.1NT1, I53-50A.1NT2, and I53-50A.1PT1 (see Supplementary Note 1 for amino acid sequences), which were designed as part of a previous effort to drive encapsulation of biomolecules in the I53-50 interior[30]. The constructs were expressed in 293F cells and the products purified via the SOSIP trimer moieties using PGT145 bNAb-affinity chromatography. Of the four variants tested, the I53-50A.1NT1 variant had the greatest yield (~0.9 mg L$^{-1}$), showed the highest proportion of trimers, and was fully cleaved (Fig. 1b and Supplementary Fig. 1a). A considerable proportion (~40%) of the fusion proteins formed aggregates during purification (Fig. 1c), but size-exclusion chromatography (SEC)-purified, aggregate-free trimer fractions were 100% native-like when imaged by negative-stain electron microscopy (NS-EM) (Fig. 1d). Furthermore, the glycosylation profiles of BG505 SOSIP-I53-50A.1NT1 and the BG505 SOSIP trimer were highly similar as determined by hydrophilic interaction ultra-performance liquid (HILIC-UPLC) (Fig. 1e). Overall, we conclude that the purified trimer components of the BG505 SOSIP-I53-50A.1NT1 fusion protein are fully cleaved and native-like.

### BG505 SOSIP-I53-50NPs assemble with high efficiency.

Purified BG505 SOSIP-I53-50A.1NT1 was then mixed with a pentameric I53-50B variant, designated I53-50B.4PT1[30], to drive assembly of the icosahedral NP immunogen. SEC purification of the SOSIP-I53-50A.1NT1 and I53-50B.4PT1 mixture revealed a peak at an elution volume of 8.5–11.5 mL, indicating formation of a high-molecular-weight complex (Fig. 2a). The SEC fractions corresponding to this peak were pooled and analyzed by reducing SDS-PAGE and NS-EM. Bands corresponding to the gp120 (~120 kDa), gp41-I53-50A.1NT1 (~48 kDa), and I53-50B.4PT1 (~18 kDa) components were all visible on the gels (Fig. 2a). When separating the peak intro three pools, pools #1 (8.5–9.5 mL) and #2 (9.5–10.5 mL) contained primarily fully assembled icosahedral NPs with a diameter of ~40–50 nm, with a minor population of partially assembled particles also visible (Fig. 2b and Supplementary Fig. 1b). Pool #3 (10.5–11.5 mL), accounting for <10% of the total peak area, contained more distorted particles and free SOSIP-I53-50A.1NT1 components (Supplementary Fig. 1b). Hence, fractions corresponding to pool #3 were not included when producing SEC-purified BG505 SOSIP-I53-50NPs.

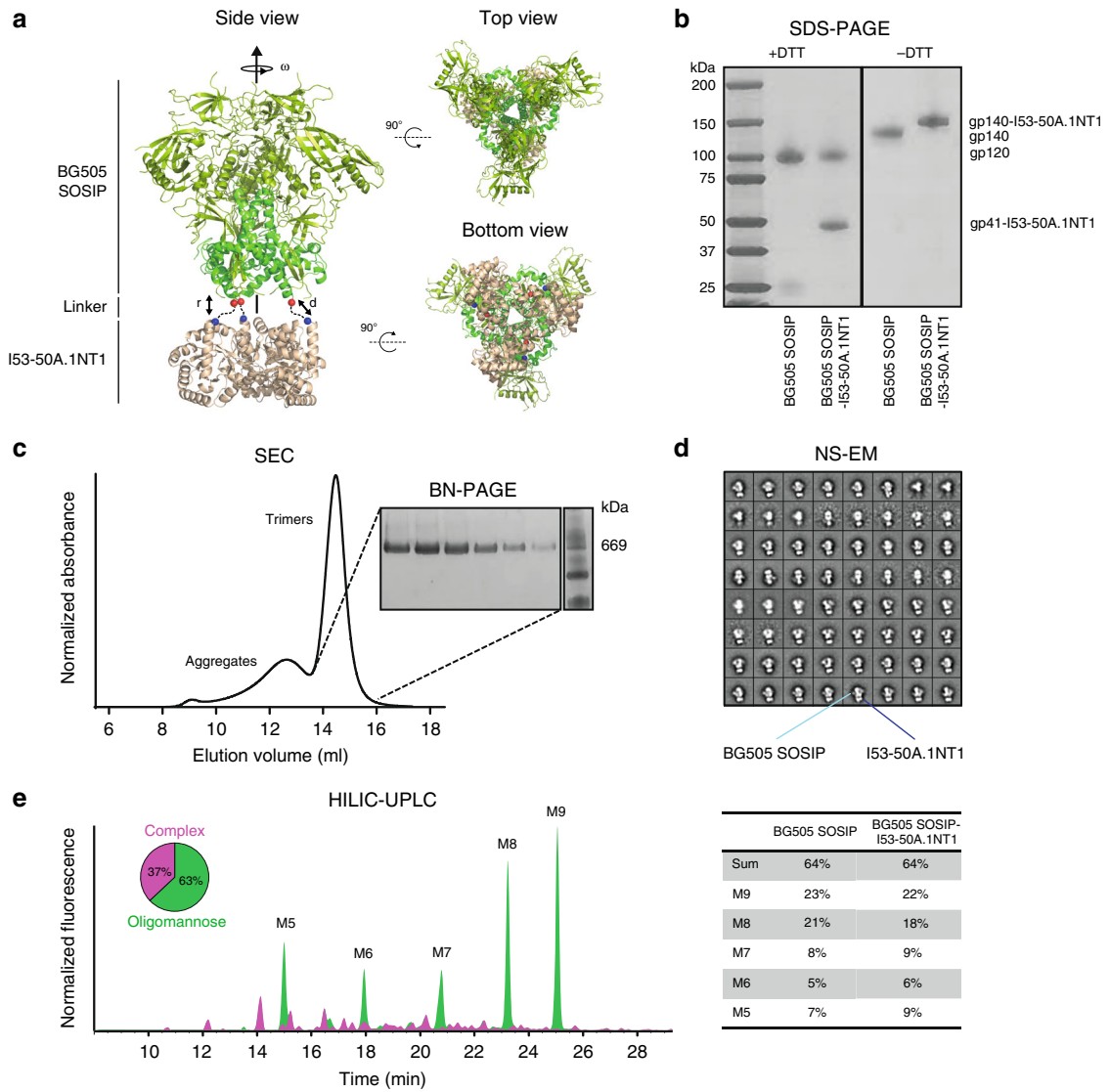

**Fig. 1** Design and biophysical properties of BG505 SOSIP-I53-50A.1NT1. **a** Schematic representation of the computational docking protocol used to identify trimeric NP components suitable for fusion to BG505 SOSIP. $r$ and $\omega$ were sampled during docking to minimize $d$ while avoiding clashes. The C termini of BG505 SOSIP and N termini of the I53-50A trimer are shown as red and blue spheres, and the dashed line indicates the linker that connects BG505 SOSIP and I53-50A. **b** Sodium dodecyl sulfate-polyacrylamide gel electrophoresis (SDS-PAGE) of BG505 SOSIP and BG505 SOSIP-I53-50A.1NT1 under reducing (left) and non-reducing (right) conditions. **c** SEC chromatogram of BG505 SOSIP-I53-50A.1NT1 with peaks corresponding to aggregates and trimers annotated (left) and blue native (BN)-PAGE of trimer fractions (right). **d** NS-EM analysis of PGT145/SEC-purified BG505 SOSIP-I53-50A.1NT1 with BG505 SOSIP (light blue) and I53-50A.1NT1 (dark blue) annotated. Shown are 2D-class averages. **e** Glycan profile of BG505 SOSIP-I53-50A.1NT1 as determined by HILIC-UPLC with complex glycans in pink and oligomannose glycans in green. The pie chart represents the percentage of total complex glycans vs. oligomannose glycans per trimer. The percentages of $Man_{5-9}GlcNAc_2$ glycans (M5–M9) are listed in the table for BG505 SOSIP and BG505 SOSIP-I53-50A.1NT1. Source data are provided as a Source Data file

We then used differential scanning fluorimetry (nanoDSF) to assess the stability of the NPs. A comparative study with the corresponding trimer showed that neither the presence of the I53-50A.1NT1 component nor presentation on the I53-50NP affected the thermostability of the Env component. Two discrete unfolding events were apparent in the thermal melting profiles for BG505 SOSIP-I53-50NPs; the SOSIP trimer components unfolded at a lower temperature than the I53-50NP core (72.5 vs. 82.0 °C, respectively; Table 1 and Supplementary Fig. 3a). Interestingly, we were unable to determine a $T_m$ for the hyperstable unmodified I53-50NP, suggesting that the presence of the SOSIP trimers may slightly decrease the thermostability of the NPs. Finally, we used Dynamic Light Scattering (DLS),

NS-EM, and SEC to show that the assembled BG505 SOSIP-I53-50NPs can withstand a freeze–thaw cycle at −80 °C (Table 1 and Supplementary Fig. 3b, c).

**High-resolution structure of BG505 SOSIP-I53-50NPs**. To further assess assembly efficiency and structural integrity, we performed single-particle cryo-EM analysis on BG505 SOSIP-I53-50NPs. After several rounds of two-dimensional (2D) and three-dimensional (3D) classification, a final set of ~3600 projections was retained and reconstructed with icosahedral symmetry to a global resolution of ~4.5 Å as determined by Fourier shell correlation (Supplementary Fig. 1c–g and Table 2). The 3D reconstruction both confirmed the predicted architecture of the I53-50

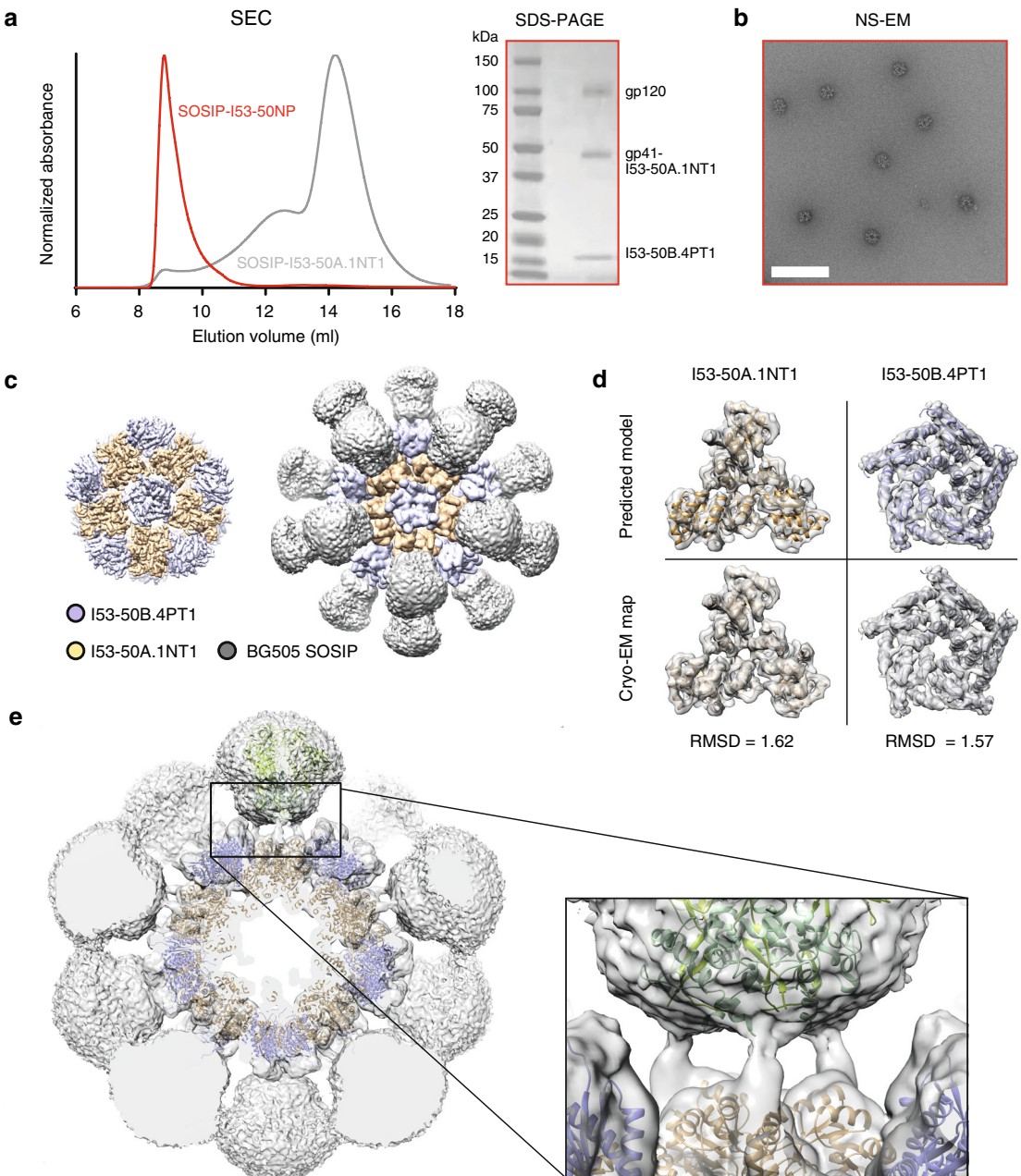

**Fig. 2** Characterization of BG505 SOSIP-I53-50NP assembly and cryo-EM structure. **a** Representative SEC chromatogram (left) of assembled BG505 SOSIP-I53-50NP (red) overlaid with BG505 SOSIP-I53-50A.1NT1 (gray) and reducing SDS-PAGE (right) of pooled BG505 SOSIP-I53-50NP fractions. **b** Raw NS-EM micrograph of pooled fractions corresponding to BG505 SOSIP-I53-50NPs. The white bar corresponds to 200 nm. **c** Segmentation of the sharpened cryo-EM map at high contour level showing details of NP core (left) and Gaussian filtered map at low contour levels showing poorly resolved SOSIP trimers (right). **d** Details of I53-50A.1NT1 (left) and I53-50B.4PT1 (right) showing the predicted model (top) and cryo-EM map (bottom). Root-mean-square deviation (RMSD) are shown below each component. **e** Low-pass-filtered cyro-EM map and pseudo-atomic model of BG505 SOSIP-I53-50NP with detail of the linker separating I53-50A.1NT1 and BG505 SOSIP. Source data are provided as a Source Data file

scaffold and the presence of BG505 SOSIP trimers attached to each I53-50A.1NT1 moiety. However, because of flexibility in the linker between the two domains (Fig. 1a), the SOSIP trimers are poorly resolved; they appear only as diffuse densities surrounding the well-resolved I53-50NP core (Fig. 2c and Supplementary Fig. 1f). At these resolutions we were able to refine an atomic model of the I53-50NP core into the cryo-EM map; the outcome revealed only minor adjustments from the computationally designed structure, primarily in the I53-50A.1NT1 moiety and at the interface between I53-50A.1NT1 and I53-50B.4PT1 (Fig. 2d).

Finally, low-pass filtering of the 4.5 Å map revealed visible density for both the BG505 SOSIP trimers and the genetic linker (Fig. 2e).

**SOSIP-I53-50NPs of other Env genotypes assemble efficiently.** To explore the generalizability of the SOSIP-I53-50NP design, we expressed I53-50A fusion proteins bearing SOSIP trimers based on the AMC011 (Clade B), ZM197M (Clade C), or ConM SOSIP (consensus of Group M) genotypes. The previously described ConM SOSIP.v7 trimer[22] could be fused to the prototypical I53-50A component and produced in good yield (~2 mg L$^{-1}$). For the

**Table 1 Morphology and thermostability of SOSIP-I53-50NPs**

| | Morphology (DLS) | | Thermostability (nanoDSF) | |
|---|---|---|---|---|
| | $R_h$ (nm)[a] | Pd (%)[b] | $T_{m1}$ (°C)[c] | $T_{m2}$ (°C)[c] |
| BG505 SOSIP | 6.7[d] | 4.4[d] | 72.4 (±1.3) | – |
| BG505 SOSIP-I53-50A.1NT1 | n.d. | n.d. | 72.6 (±0.6) | – |
| BG505 SOSIP-I53-50NP | 22.6 | 5.2 | 72.5 (±0.2) | 82.0 (±0.2) |
| ConM SOSIP | 6.4[e] | 12.5[e] | 65.7 (±0.1) | – |
| ConM SOSIP-I53-50A | n.d. | n.d. | 67.8 (±0.1) | – |
| ConM SOSIP-I53-50NP | 22.6 | 7.1 | 69.1 (±0.1) | 84.4 (±0.2)[f] |
| ZM197M SOSIP-I53-50NP | 24.6 | 13.8 | 72.1 (±0.1) | 83.9 (±0.6)[f] |
| AMC011 SOSIP-I53-50NP | 23.8 | 11.0 | 74.0 (±0.4) | 89.0 (±1.3)[f] |

All samples have been subjected to a freeze–thaw cycle at −80 °C prior to experiments. All SOSIP-I53-50NPs have the expected $R_h$ of an intact NP and are monodisperse after a freeze–thaw cycle at −80 °C. For BG505 and ConM, corresponding SOSIP trimers and SOSIP fusion proteins are included for comparison. The AMC011 SOSIP-I53-50NP solution was mixed 3:1 with Protein Stabilizing Cocktail (ThermoScientific) prior to freezing. NanoDSF experiments were performed three times. Shown are the means of three measurements with SDs between brackets. See also Supplementary Fig. 3.
*n.d.* not determined
[a]$R_h$ = hydrodynamic radius
[b]Pd = polydispersity, a sample with a Pd < 14% is considered monodisperse
[c]$T_m$ = melting temperature
[d]Values adopted from ref. [36]
[e]Values adopted from ref. [22]
[f]As the NanoDSF software did not give a clean valley in the first derivative, it was unable to determine the $T_m$. Therefore, the $T_m$ was determined manually by taking the lowest point in the first derivative

**Table 2 Cryo-EM data collection, refinement, and validation statistics**

| | BG505 SOSIP-I53-50NP (EMDB-20261) (PDB 6P6F) |
|---|---|
| **Data collection and processing** | |
| Magnification | 38,168 |
| Voltage (kV) | 300 |
| Electron exposure (e−/Å²) | 46.6 |
| Defocus range (μm) | 1.2–5.1 |
| Pixel size (Å) | 1.31 |
| Symmetry imposed | I1 |
| Initial particle images (no.) | 3996 |
| Final particle images (no.) | 3590 |
| Map resolution (Å) | 4.5 (0.143) |
| FSC threshold | |
| Map resolution range (Å) | 4.4–6 |
| **Refinement** | |
| Initial model used (PDB code) | NA |
| Model resolution (Å) | NA |
| FSC threshold | |
| Model resolution range (Å) | NA |
| Map sharpening B factor (Å²) | −149 |
| Model composition | |
| Non-hydrogen atoms | 0 |
| Protein residues | 351 |
| Ligands | 0 |
| B factors (Å²) | NA |
| Protein | |
| Ligand | |
| R.m.s. deviations | |
| Bond lengths (Å) | 0.017 |
| Bond angles (°) | 1.586 |
| Validation | |
| MolProbity score | 0.5 |
| Clashscore | 0 |
| Poor rotamers (%) | 0 |
| EMRinger score | 1.55 |
| Ramachandran plot | |
| Favored (%) | 99.7 |
| Allowed (%) | 0 |
| Disallowed (%) | 0 |

clade B and C constructs, the AMC011 SOSIP.v8.1 and ZM197M SOSIP.v5.2(519A, 568D, 570H, 585H) trimers, respectively, were also fused to the prototypical I53-50A. These trimers have additional modifications compared with previously described versions[37,38], as outlined in Supplementary Table 1, which decrease non-NAb reactivity and, particularly for ZM197M SOSIP, increase yields (Supplementary Fig. 2a-c)[26,36,39]. As seen with BG505 SOSIP-I53-50A.1NT1, the three new fusion proteins could be purified as native-like and efficiently cleaved trimers (Supplementary Fig. 2d, e). These constructs are hereafter referred to as ConM, ZM197M, and AMC011 SOSIP-I53-50A for convenience.

SEC of ConM, ZM197M, or AMC011 SOSIP-I53-50A mixed with I53-50B.4PT1 resulted in a peak at the same elution volume as BG505 SOSIP-I53-50NP (Fig. 3a). Similar to BG505 SOSIP-I53-50NPs, the majority of ConM and ZM197M SOSIP-I53-50NPs appeared as well-assembled particles in NS-EM images after a single freeze–thaw cycle and these NPs were monodisperse and thermostable as determined by DLS and DSF, respectively (Fig. 3b, Supplementary Fig. 3a, b, and Table 1). SEC was also performed to further demonstrate that the ConM SOSIP-I53-50NPs are able to withstand a freeze–thaw cycle at −80 °C (Supplementary Fig. 3c). AMC011 SOSIP-I53-50NPs, however, required a cryoprotectant, either sucrose or the Protein Stabilizing Cocktail from ThermoScientific, to maintain its integrity during freeze–thawing (Fig. 3b and Supplementary Fig. 3d).

**SOSIP trimers on I53-50NPs maintain their antigenicity**. We used surface plasmon resonance (SPR) to determine whether the antigenicity of the SOSIP trimers was altered by their presentation on the surface of the I53-50NPs. BG505 or ConM SOSIP trimers and the corresponding SOSIP-I53-50NPs were captured onto chips by their His-tags (for SOSIP trimers on the C terminus, for SOSIP-I53-50NPs on the exterior-facing C terminus of the I53-50B.4PT1 pentamer) and the association and dissociation of various bNAbs and non-NAbs was monitored. The PGT145 and PG9 bNAbs against trimer-dependent apical epitopes bound similarly to the SOSIP trimers in the two contexts (Supplementary Fig. 4a). However, bNAbs directed to epitopes further down from the apex bound less well to SOSIP-I53-50NP than the corresponding trimers, and this difference increased when the epitope was located closer to the base. Various non-NAbs against

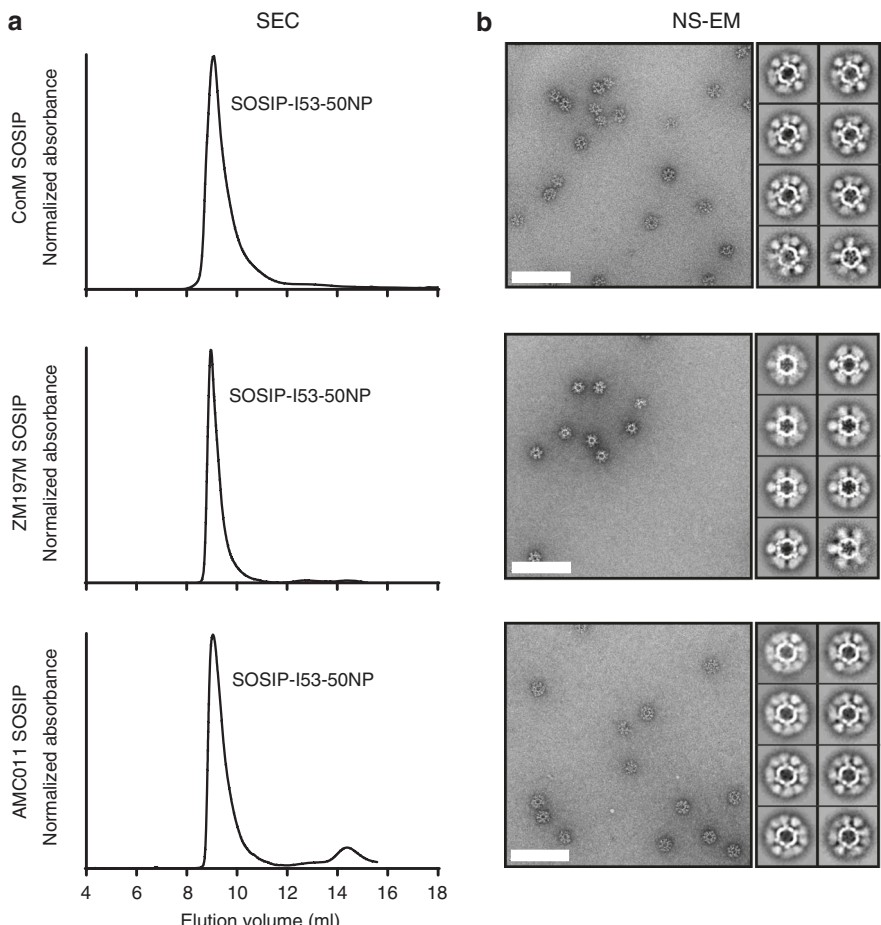

**Fig. 3** Characterization of ConM, ZM197M, and AMC011 SOSIP-I53-50NP assembly. **a** SEC chromatograms of assembled ConM, ZM197, and AMC011 SOSIP-I53-50NPs (from top to bottom). **b** Raw NS-EM micrographs and 2D-class averages of the pooled SOSIP-I53-50NP fractions corresponding to the SEC chromatogram to the direct left. Particles were subjected to a freeze–thaw cycle prior to NS-EM analysis. The AMC011 SOSIP-I53-50NP solution was mixed 3:1 with Protein Stabilizing Cocktail (ThermoScientific) prior to freezing. The white bar corresponds to 200 nm. Source data are provided as a Source Data file

the CD4bs (b6), the CD4-induced epitope (17b), and the gp120-V3 region (14e, 19b, and 39F) were similarly non-reactive with the BG505 and ConM SOSIP trimers alone and on I53-50NPs (Supplementary Fig. 4a). We observed similarly strong binding by bNAbs and weak or absent binding by non-NAbs to ZM197M and AMC011 SOSIP alone and on I53-50NPs (Supplementary Fig. 4b).

Fabs of seven bNAbs to different epitope clusters bound with similar kinetics and stoichiometry to the trimers presented in the two contexts, with small variations in both directions (Table 3 and Supplementary Figs. 4c and 5b). Specifically, the stoichiometric values for PGT145, PGT151, and 3BNC117 Fabs, close to 1, 2, and 3 per trimer, respectively, confirmed that the SOSIP-I53-50NPs present 20 antigenically intact trimers per particle (Table 3). Comparing IgGs and Fabs suggested that the lower binding by some IgGs to SOSIP-I53-50NPs is a result of bivalent binding by IgG to its epitope (such as for PGT130, PGT151, and ACS202) or steric clashes due to the bulk of the IgG (see Supplementary Fig. 4 for further analyses and explanations). Overall, these findings demonstrate that SOSIP trimers maintain their native-like trimeric antigenicity when presented on I53-50NPs.

To investigate avidity and accessibility effects of the multivalent display of trimers on I53-50NPs, IgG mAbs were immobilized on the chip. We observed a high binding signal for BG505 and

ConM SOSIP-I53-50NPs to the bNAbs VRC01 (CD4bs), PGT122 (V3-base and -glycan), and PGT145 (apex) in comparison with the corresponding SOSIP trimers. In contrast, the interface bNAbs PGT151 and VRC34.01, and the gp41-directed 3BC315, all displayed a lower binding signal to SOSIP-I53-50NPs than SOSIP trimers (Fig. 4a and Supplementary Fig. 5c). However, it is hard to draw conclusions on the binding signal alone, as the greater mass of the SOSIP-I53-50NPs magnifies the signal caused by each binding event (see Supplementary Methods for further explanation). Thus, to obtain a more quantitative measure of bNAb binding and accesibility, the number of molecules of SOSIP trimer or NPs binding to the bNAbs was calculated from an experiment with 2 nM trimers or particles (see Supplementary Methods for calculation). Whereas a similar number of molecules bound to PGT145, PGT122, and VRC01, markedly fewer molecules of SOSIP-I53-50NP than SOSIP trimers bound to VRC34.01, PGT151, and 3BC315 (Fig. 4b). Finally, we also observed a markedly lower binding of BG505 SOSIP-I53-50NP than their corresponding trimers to BG505-specific mAbs targeting the immunodominant 241/289 glycan hole (RM19P, RM19A1, RM20F; Cottrell et al., manuscript in preparation) and trimer base (12N) (Supplementary Fig. 5a)[40]. Thus, bNAb epitopes on the SOSIP-I53-50NPs from the apex down to the CD4bs are highly accessible for multivalent interactions, but this

**Table 3 Kinetics and stoichiometry of Fab binding to SOSIP trimers and SOSIP-I53-50NPs**

Langmuir modeling of Fabs binding

| Fab | Envelope protein | n | $k_{on}$ (Ms$^{-1}$) | $k_{off}$ (s$^{-1}$) | $K_D$ (nM) | $S_m$/NP | $S_m$/trimer |
|---|---|---|---|---|---|---|---|
| PGT145 | BG505 SOSIP | 3 | $2.4 \times 10^5 \pm 5.9 \times 10^4$ | $3.5 \times 10^{-4} \pm 3.3 \times 10^{-6}$ | $1.8 \pm 0.62$ | – | $0.94 \pm 1.9 \times 10^{-2}$ |
| | BG505 SOSIP-I53-50NP | 2 | $2.1 \times 10^5 \pm 1.6 \times 10^5$ | $3.1 \times 10^{-4} \pm 1.2 \times 10^{-4}$ | $4.8 \pm 4.2$ | $18 \pm 0.64$ | $0.88 \pm 3.2 \times 10^{-2}$ |
| | ConM SOSIP | 4 | $1.6 \times 10^5 \pm 1.1 \times 10^4$ | $3.8 \times 10^{-4} \pm 9.5 \times 10^{-6}$ | $2.5 \pm 0.15$ | – | $0.79 \pm 5.2 \times 10^{-2}$ |
| | ConM SOSIP-I53-50NP | 3 | $1.5 \times 10^5 \pm 9.5 \times 10^3$ | $4.2 \times 10^{-4} \pm 3.8 \times 10^{-5}$ | $2.8 \pm 0.26$ | $20 \pm 2.5$ | $0.94 \pm 9.9 \times 10^{-2}$ |
| 3BNC117 | BG505 SOSIP | 3 | $3.5 \times 10^4 \pm 1.2 \times 10^3$ | $9.0 \times 10^{-5} \pm 1.3 \times 10^{-5}$ | $2.6 \pm 0.33$ | – | $2.9 \pm 0.18$ |
| | BG505 SOSIP-I53-50NP | 2 | $2.0 \times 10^4 \pm 5.0 \times 10^2$ | $4.0 \times 10^{-5} \pm 3.5 \times 10^{-6}$ | $2.1 \pm 0.15$ | $55 \pm 0.89$ | $2.8 \pm 4.5 \times 10^{-2}$ |
| | ConM SOSIP | 2 | $2.7 \times 10^4 \pm 5.0 \times 10^2$ | $2.6 \times 10^{-4} \pm 5.0 \times 10^{-6}$ | $9.6 \pm 0$ | – | $2.8 \pm 0$ |
| | ConM SOSIP-I53-50NP | 2 | $1.4 \times 10^4 \pm 5.0 \times 10^2$ | $7.6 \times 10^{-5} \pm 2.4 \times 10^{-5}$ | $5.8 \pm 1.8$ | $50 \pm 0.39$ | $2.5 \pm 1.9 \times 10^{-2}$ |
| PGT122 | BG505 SOSIP | 3 | $1.3 \times 10^4 \pm 6.7 \times 10^2$ | $1.6 \times 10^{-4} \pm 3.1 \times 10^{-5}$ | $13 \pm 2.7$ | – | $1.3 \pm 1.1 \times 10^{-2}$ |
| | BG505 SOSIP-I53-50NP | 3 | $6.6 \times 10^3 \pm 2.0 \times 10^2$ | $2.4 \times 10^{-4} \pm 8.4 \times 10^{-5}$ | $36 \pm 13$ | $25 \pm 1.2$ | $1.3 \pm 6.2 \times 10^{-2}$ |
| | ConM SOSIP | 5 | $4.2 \times 10^4 \pm 1.4 \times 10^3$ | $4.2 \times 10^{-5} \pm 6.3 \times 10^{-6}$ | $0.99 \pm 0.14$ | – | $2.8 \pm 4.7 \times 10^{-2}$ |
| | ConM SOSIP-I53-50NP | 6 | $3.5 \times 10^4 \pm 8.8 \times 10^2$ | $1.0 \times 10^{-4} \pm 1.7 \times 10^{-5}$ | $2.9 \pm 0.49$ | $51 \pm 0.82$ | $2.5 \pm 4.1 \times 10^{-2}$ |
| 35O22 | BG505 SOSIP | 3 | $1.2 \times 10^4 \pm 3.3 \times 10^2$ | $2.6 \times 10^{-4} \pm 2.7 \times 10^{-5}$ | $22 \pm 2.3$ | – | $1.5 \pm 5.5 \times 10^{-2}$ |
| | BG505 SOSIP-I53-50NP | 2 | $4.6 \times 10^3 \pm 5.0 \times 10^2$ | $2.4 \times 10^{-4} \pm 1.5 \times 10^{-5}$ | $51 \pm 9.0$ | $25 \pm 1.9$ | $1.2 \pm 0.10$ |
| | ConM SOSIP | 2 | $2.3 \times 10^4 \pm 5.0 \times 10^2$ | $8.0 \times 10^{-5} \pm 1.9 \times 10^{-5}$ | $3.5 \pm 0.90$ | – | $1.1 \pm 4.8 \times 10^{-2}$ |
| | ConM SOSIP-I53-50NP | 2 | $5.5 \times 10^4 \pm 5.0 \times 10^2$ | $1.0 \times 10^{-4} \pm 1.1 \times 10^{-5}$ | $1.9 \pm 0.25$ | $28 \pm 1.3$ | $1.4 \pm 6.5 \times 10^{-2}$ |
| 3BNC315 | BG505 SOSIP | 3 | $5.5 \times 10^3 \pm 2.7 \times 10^2$ | $4.1 \times 10^{-5} \pm 2.3 \times 10^{-5}$ | $8.8 \pm 4.9$ | – | $1.7 \pm 4.5 \times 10^{-2}$ |
| | BG505 SOSIP-I53-50NP | 2 | $4.1 \times 10^3 \pm 0$ | $<10^{-5}$ | $<3$ | $22 \pm 0.52$ | $1.1 \pm 2.6 \times 10^{-2}$ |
| | ConM SOSIP | 2 | $3.6 \times 10^4 \pm 4.5 \times 10^3$ | $1.8 \times 10^{-4} \pm 2.0 \times 10^{-5}$ | $5.1 \pm 0.15$ | – | $2.2 \pm 0.19$ |
| | ConM SOSIP-I53-50NP | 2 | $1.5 \times 10^4 \pm 1.0 \times 10^3$ | $6.0 \times 10^{-5} \pm 2.4 \times 10^{-5}$ | $4.2 \pm 2.0$ | $33 \pm 1.7$ | $1.6 \pm 8.3 \times 10^{-2}$ |

Conformational-change modeling of PGT151 Fab binding

| Fab | Envelope protein | n | $k_{on}$ (Ms$^{-1}$) | $k_{off}$ (s$^{-1}$) | $k_f$ (s$^{-1}$) | $k_b$ (s$^{-1}$) | $K_D$ (nM) |
|---|---|---|---|---|---|---|---|
| PGT151 | BG505 SOSIP | 2 | $9.8 \times 10^4 \pm 5.0 \times 10^2$ | $1.4 \times 10^{-3} \pm 8.6 \times 10^{-4}$ | $6.2 \times 10^{-3} \pm 1.1 \times 10^{-3}$ | $1.3 \times 10^{-3} \pm 3.7 \times 10^{-4}$ | $15 \pm 5.8$ |
| | BG505 SOSIP-I53-50NP | 2 | $5.4 \times 10^4 \pm 1.0 \times 10^3$ | $2.8 \times 10^{-4} \pm 5.0 \times 10^{-6}$ | $8.6 \times 10^{-5} \pm 8.4 \times 10^{-5}$ | $9.1 \times 10^{-4} \pm 8.9 \times 10^{-4}$ | $5.1 \pm 0.19$ |
| | ConM SOSIP | 3 | $9.3 \times 10^4 \pm 1.5 \times 10^3$ | $6.2 \times 10^{-3} \pm 3.1 \times 10^{-4}$ | $5.7 \times 10^{-3} \pm 2.7 \times 10^{-4}$ | $3.3 \times 10^{-4} \pm 5.4 \times 10^{-5}$ | $67 \pm 2.3$ |
| | ConM SOSIP-I53-50NP | 2 | $4.7 \times 10^4 \pm 1.0 \times 10^3$ | $1.2 \times 10^{-2} \pm 5.0 \times 10^{-4}$ | $1.2 \times 10^{-2} \pm 0$ | $2.1 \times 10^{-4} \pm 2.5 \times 10^{-5}$ | $250 \pm 5.4$ |

| Fab | Envelope protein | n | $K_F$ | $K_{D(conf)}$ (nM) | $S_m$/NP | $S_m$/trimer |
|---|---|---|---|---|---|---|
| PGT151 | BG505 SOSIP | 2 | $5.2 \pm 2.2$ | $2.3 \pm 0.11$ | – | $1.8 \pm 5.6 \times 10^{-3}$ |
| | BG505 SOSIP-I53-50NP | 2 | $0.17 \pm 7.4 \times 10^{-2}$ | $4.4 \pm 0.19$ | $29 \pm 0.17$ | $1.5 \pm 8.4 \times 10^{-3}$ |
| | ConM SOSIP | 3 | $18 \pm 4.0$ | $3.7 \pm 0.56$ | – | $1.8 \pm 0.12$ |
| | ConM SOSIP-I53-50NP | 2 | $59 \pm 7.2$ | $3.8 \pm 0.36$ | $37 \pm 0.55$ | $1.9 \pm 2.8 \times 10^{-2}$ |

Shown are average kinetics ($k_{on}$ = the on-rate constant, $k_{on}$ = the off-rate constant, $K_D$ = disassociation constant) and stoichiometry values ($S_m$ = stoichiometry) of replicate (n) SPR experiments with standard error of the mean. Since a Langmuir model did not fit well for PGT151 Fab binding to ConM SOSIP a conformational-change model was used for this particular Fab. Average kinetics and stoichiometry values are shown in addition to conversion constants $k_f$ and $k_b$ and the forward-conversion-equilibrium constant $K_F$. See also Supplementary Fig. 4 and Supplementary Methods for further analysis and explanation

is not the case for some epitopes further down the trimer near gp41 and epitopes at the trimer base.

**SOSIP-I53-50NPs increase B-cell activation in vitro.** Next, we compared the abilities of equimolar BG505 or ConM SOSIP trimers and their SOSIP-I53-50NP counterparts to activate Env-specific B cells in vitro. In control experiments, unmodified I53-50NPs lacking the SOSIP trimers had no effect on any of the B-cell lines. The corresponding BG505 and ConM SOSIP trimers did not activate the VRC01- or PGT145-expressing B cells, at the concentration used, but an equimolar amount of trimers presented on I53-50NPs induced $Ca^{2+}$ flux signals very efficiently (Fig. 4c). For the PGT121 cell line, the BG505 and ConM SOSIP-I53-50NPs were again more efficient activators than the trimers, although the differences were less dramatic than for the other two lines (Fig. 4c). In addition, ConM SOSIP-I53-50NPs activated the B cells more efficiently than ConM SOSIP-ferritin (homomeric NPs presenting eight SOSIP trimers[22]), consistent with a previous study showing a positive correlation between SOSIP density and B-cell activation[27].

**Immunogenicity of ConM and BG505 SOSIP-I53-50NPs in rabbits.** ConM and BG505 SOSIP trimers elicit autologous NAbs that target different immunodominant sites when they are used as immunogens. In BG505 SOSIP trimer-immunized rabbits, the most commonly immunogenic NAb epitope involves a hole in the glycan shield at positions 241 and 289. This glycan hole epitope is located nearer the base of the SOSIP trimer than its apex[40,41]. In contrast, ConM SOSIP trimer-immunized rabbits generate autologous NAbs that primarily target a V1V2 epitope near the apex of the trimer[22]. For AMC011 and ZM197M, it remains unknown where the immunodominant epitope is located. By selecting ConM and BG505 SOSIP-I53-50NPs for immunization, we aimed to evaluate whether immunodominant epitope location affects the performance of SOSIP-I53-50NPs as immunogens. Rabbits were immunized at weeks 0, 4, and 20 with 30 µg of BG505 or ConM SOSIP, or the equimolar amount presented on I53-50NPs, formulated in GLA-LSQ adjuvant (liposomes containing glucopyranosyl lipid adjuvant (GLA) and saponin (QS21)), which we found did not influence trimer or NP integrity (Fig. 5a, f and Supplementary Fig. 6a). BG505 immunogen recipient animals were given an additional vaccination at week 8 (Fig. 5f). To directly compare the two-component NP platform with an existing homomeric NP platform, one group of rabbits received ConM SOSIP-ferritin NPs (Fig. 5a).

In comparison with ConM SOSIP-ferritin, rabbits immunized with ConM SOSIP-I53-50NP induced ConM SOSIP-binding titers that were significantly lower at week 2 but higher at week 8 and 20. ConM SOSIP trimers also induced significantly higher binding titers than ConM SOSIP-ferritin at week 20. We did not observe significant differences in binding titers when comparing ConM SOSIP trimers and ConM SOSIP-I53-50NPs (Fig. 5b). However, ConM SOSIP-I53-50NPs induced significantly higher NAb titers against autologous ConM virus than the corresponding trimer at all time points (Fig. 5c, d and Supplementary Table 2). In particular, the ConM SOSIP-I53-50NPs induced

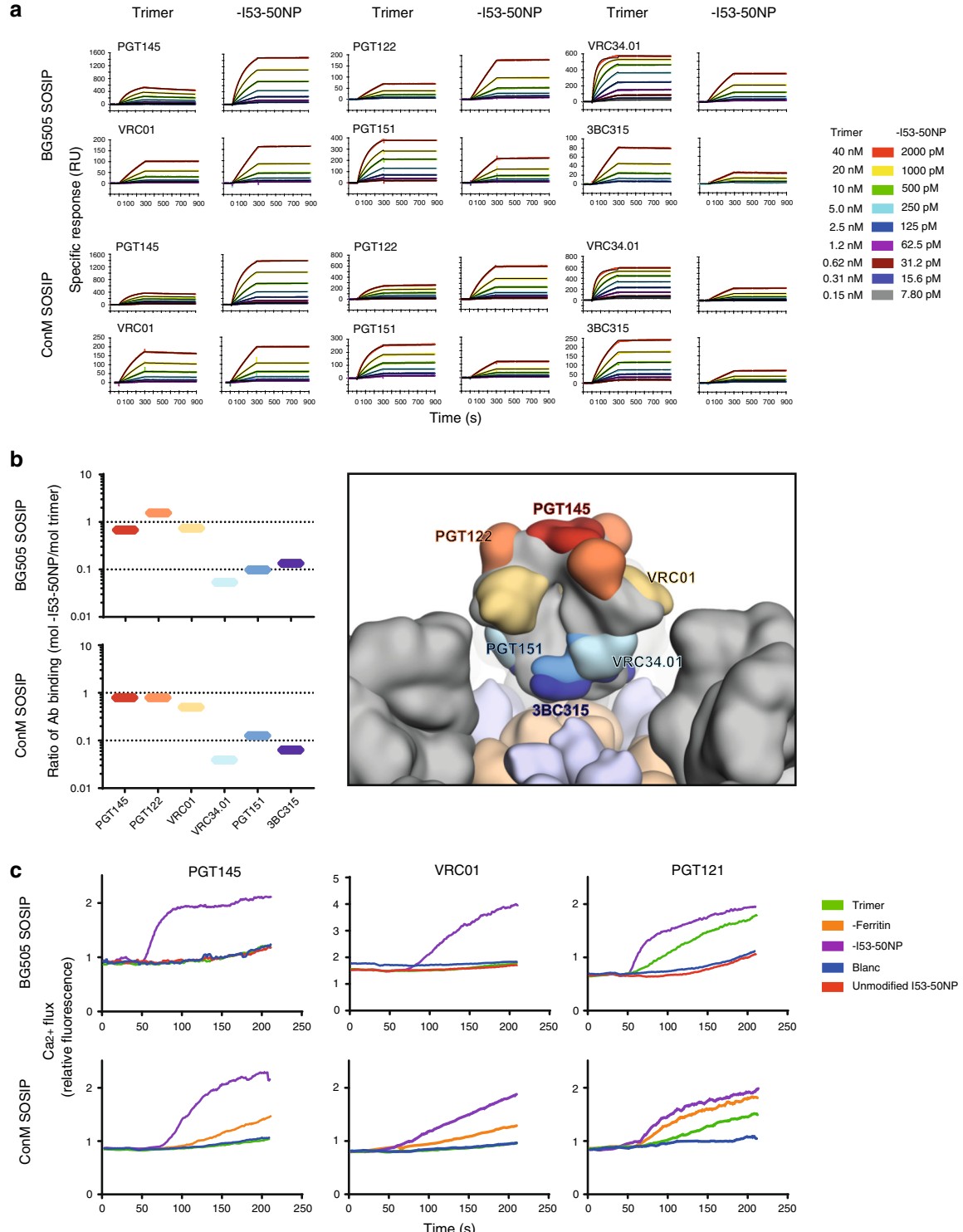

**Fig. 4** SPR and B-cell activation analysis of BG505 and ConM SOSIP-I53-50NP. **a** Each pair of sensorgrams shows the binding of SOSIP trimer (left) and SOSIP-I53-50NPs (right) to immobilized IgG of the bNAbs indicated above the sensorgrams. Trimers and NPs were titrated at a constant ratio of concentrations corresponding to equal amounts of SOSIP per volume as indicated in the color code to the right. Langmuir model fits in black are overlaid on the colored binding curves for each concentration. Please note that although Langmuir curves were fitted, no kinetic modeling was performed due to severe mass-transport limitations (for further explanation, see SOSIP trimer and SOSIP-I53-50NP binding to immobilized mAbs by SPR in Supplementary Methods). **b** The difference in macromolecules of SOSIP trimer or SOSIP-I53-50NP bound to several immobilized bNAbs plotted as a ratio (left) and a model of SOSIP (gray) presented on the I53-50NP (I53-50A in beige, I53-50B.4PT1 in light blue) with the corresponding bNAb footprints shown. **c** Activation of PGT145, VRC01, and PGT121-expressing B cells, measured by $Ca^{2+}$ flux over 210 seconds for BG505 and ConM SOSIP-I53-50NPs (purple), ConM SOSIP-ferritin (orange), and the corresponding trimer (green). Unmodified I53-50NPs (red) were included to determine nonspecific activation by the I53-50NP core. Blanc, for which PBS was used, functioned as an additional negative control (blue). Source data are provided as a Source Data file

remarkably higher NAb responses after one immunization (~41-fold), but the differences between the NP and trimer groups became less pronounced after multiple immunizations (~7- and ~4-fold after the first and second boost, respectively), suggesting that the NPs are particularly effective at priming a NAb response. ConM SOSIP-I53-50NP also induced higher median autologous NAb titers than ConM SOSIP-ferritin at week 6, week 8, and week 12. At weeks 4, 20, and 22, we did not observe a significant difference in median autologous NAb titers, although particularly at week 4 and 20 a trend towards higher NAb titers was observed for ConM SOSIP-I53-50NP recipients (Fig. 5c, d and Supplementary Table 2). The ConM virus is a neutralization-sensitive Tier-1A virus, but ConM SOSIP has been shown to also induce NAbs against the sequence-related ConS virus[22]. ConM SOSIP-I53-50NP induced higher median ConS NAb titers than trimers at week 6 but no significant differences were observed at week 4 or 22 (Fig. 5d and Supplementary Table 2). The ConS NAb titers elicited by ConM SOSIP-I53-50NP also tended to be higher than those elicited by ConM SOSIP-ferritin at week 6 and 22, although the differences were not statistically significant.

We assessed the ability of the rabbit sera to neutralize several Tier-1A viruses, which can be indicative of the elicitation of non-NAbs (for Tier-2 viruses) that target the V3 loop, a cluster of epitopes that is inaccessible on closed, native-like trimers[7]. NAb titers against SF162, MW965, and MN.3 were relatively low and very similar for sera from animals that received SOSIP-I53-50NPs or SOSIP trimers. In contrast, SOSIP-ferritin NPs induced significantly higher NAb responses against SF162 and MW965 than did SOSIP-I53-50NPs, which we suggest represents a response to a proportion of misfolded trimers on the SOSIP-ferritin NPs (Fig. 5e). A similar but less pronounced trend was observed with MN.3. We did not detect strong NAb responses against heterologous Tier-2 viruses (Supplementary Table 2).

Whereas presentation of ConM SOSIP trimers on I53-50NPs clearly enhanced their immunogenicity, this was not the case for BG505 SOSIP trimers. Rabbits immunized with BG505 SOSIP-I53-50NPs induced significantly lower binding Ab titers to the corresponding trimer in comparison with BG505 SOSIP recipients at all time points, except week 20. In addition, BG505 SOSIP-I53-50NPs showed a trend towards poorer autologous NAb responses at all time points, although the differences were not statistically significant (Fig. 5h and Supplementary Table 3).

To test whether the I53-50 core of the SOSIP-I53-50NPs was immunogenic, we measured Ab binding titers against I53-50NPs devoid of SOSIP trimers. Both ConM and BG505 SOSIP-I53-50NPs elicited strong binding Ab responses after the prime that were boosted by subsequent immunizations (Supplementary Fig. 6b). However, in comparison with SOSIP trimer-binding titers, the I53-50NP-binding titers increased significantly less after the boost immunizations, suggesting that primarily SOSIP-specific responses are boosted after subsequent SOSIP-I53-50NP immunizations (Supplementary Fig. 6c).

**Epitope location influences the immunogenicity of SOSIP-I53-50NPs.** The serological data, viewed in the context of the antigenic characterization by SPR, suggested that restricted accessibility to the immunodominant 241/289 glycan hole on BG505 SOSIP-I53-50NPs might explain the inability of these NPs to enhance autologous NAb responses. To test this hypothesis, we used neutralization assays with mutant viruses and depletion reagents, as well as competition ELISAs to compare the epitope specificities of the polyclonal responses induced by the BG505 and ConM immunogens.

Knocking in both glycans at positions 241 and 289 in the BG505 virus dramatically reduced the neutralization sensitivity to sera from BG505 SOSIP trimers, but had significantly less effect when sera from BG505 SOSIP-I53-50NP recipient animals were used. Similar trends were apparent when the single glycan knock-in viruses were used (Fig. 6a, b and Supplementary Table 3). BG505 SOSIP trimers are also known to induce NAb responses against the area involving the N611 glycan, located on gp41[40,41]. Removing the N611 glycan affected BG505 neutralization in six out of the eight animal immunized with SOSIP trimers. However, this had almost no effect for sera from the BG505 SOSIP-I53-50NP group (Fig. 6b and Supplementary Table 3). These results indicate that presentation of BG505 SOSIP on I53-50NPs reduces NAb responses directed against the region surrounding the 241/289 glycan hole as well as the 611 glycan, consistent with the reduced accessibility of this epitope we observed using SPR. In this context, one might expect that the Ab response would shift towards more apex-proximal epitopes. However, although neutralization by two rabbits (2171 and 2176) in the BG505 SOSIP-I53-50NP group showed a distinctly strong dependency on the previously described C3 region/residue 465 epitope[41], we observed a significantly lower dependency on the V1V2 region for the BG505 SOSIP-I53-50NP than for the BG505 SOSIP trimer recipients (Supplementary Table 3). Thus, no clear sign of immune-focusing to apex-proximal epitopes was observed concomitant with the lower 241/289 glycan hole immunogenicity.

To corroborate the findings from the neutralization assays with viral mutants, we performed neutralization depletion experiments. Sera were pre-incubated either with BG505 SOSIP or BG505 SOSIP with glycans knocked in at positions 241 and 289, prior to measuring the autologous neutralization. Pre-incubation with BG505 SOSIP blocked BG505 neutralization in all cases. However, incubation with BG505 SOSIP that had the 241 and 289 glycan knocked in had almost no effect on the neutralization in the sera of the BG505 SOSIP trimer recipients, with the exception of rabbit 2170, which showed a strong NAb response towards the V1V2 region (Supplementary Fig. 7a). In contrast, only one animal from the BG505 SOSIP-I53-50NP group showed strong NAb responses directed to the 241/289 glycan hole, as illustrated by the loss in neutralization depletion when the serum was pre-incubated with BG505 SOSIP with glycans knocked in at positions 241 and 289. All other BG505 SOSIP-I53-50NP recipients either did not develop strong BG505 neutralization or developed BG505 neutralization that was not dramatically affected by the absence or presence of the 241/289 glycan hole.

To further gauge the differences between the immunogenicity of epitopes on BG505 SOSIP trimers and NPs, we performed competition ELISAs using bNAbs with specificities ranging from the trimer apex to the trimer gp120-gp41 interface. Sera from BG505 SOSIP-I53-50NPs recipients competed significantly less efficiently than BG505 trimer recipients with bNAbs 8ANC195, ACS202, and 35O22 targeting the gp120-gp41 interface near the 241/289 glycan hole (Fig. 6a, c), and this competition correlated significantly with BG505 neutralization (Supplementary Fig. 7b, left). In contrast, we observed no significant difference in competition of sera from BG505 SOSIP trimer or BG505 SOSIP-I53-50NP recipients to apex-targeting bNAbs (CH01, PGT128, and VRC01) (Fig. 6c). Nor did we observe a significant correlation between serum competition and BG505 neutralization (Supplementary Fig. 7b, right). Furthermore, we performed a competition ELISA with RM19R, a non-NAb targeting the highly immunodominant peptidic epitope on the exposed base of the trimer (Cottrell et al., manuscript in preparation)[42,43] RM19R competed significantly less with sera from BG505 SOSIP-I53-50NP recipients than sera from rabbits that received BG505 SOSIP, indicating that presenting SOSIPs on I53-50NPs reduces the immunogenicity of the trimer base (Fig. 6c).

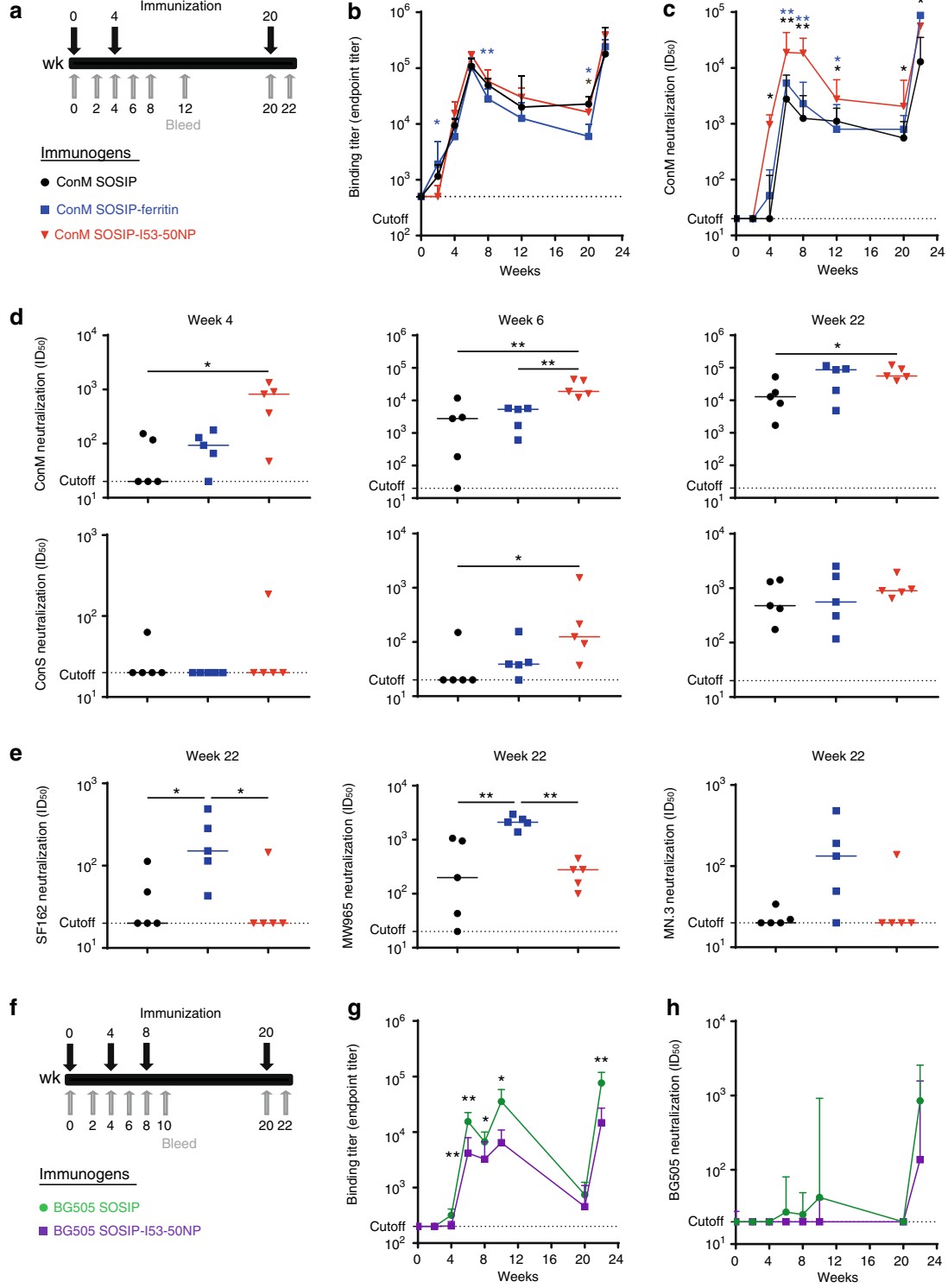

**Fig. 5** Immunogenicity of ConM and BG505 SOSIP-I53-50NPs. **a**, **f** Schematic representation of the immunization schedule and color coding for the immunogens tested. **b**–**e**, **g**, **h** Statistical differences between two groups ($n = 5$ individual rabbits for (**b**–**e**); $n = 8$ individual rabbits for (**g**, **h**)) were determined using unpaired two-tailed Mann–Whitney $U$-tests (*$p < 0.05$; **$p < 0.01$). **b**. ConM SOSIP endpoint binding titers over time as measured by Ni-NTA enzyme-linked immunosorbent assay (ELISA). Blue stars indicate significant differences for ConM SOSIP-ferritin vs. ConM SOSIP-I53-50NP. Brown stars indicate significant differences for ConM SOSIP vs. ConM SOSIP-ferritin. Shown are medians with interquartile range. **c** ConM midpoint neutralization titers over time. Blue and black stars indicate significant differences for ConM SOSIP-ferritin vs. ConM SOSIP-I53-50NP and ConM SOSIP vs. ConM SOSIP-I53-50NP, respectively. Shown are medians with interquartile range. **d** ConM and ConS midpoint neutralization titers at week 4 (left), week 6 (middle), and week 22 (right). Horizontal bars indicate the median. For replicate neutralization data from Duke UMC, see also Supplementary Table 2. **e** SF162, MW965, and MN.3 midpoint neutralization titers at week 22. Horizontal bars indicate the median. See also Supplementary Table 2. **g** BG505 SOSIP endpoint binding titers over time as measured by Ni-NTA ELISA. Shown are medians with interquartile range. **h** BG505 midpoint neutralization titers over time. Shown are medians with interquartile range. See also Supplementary Table 3. Source data are provided as a Source Data file

To map the immunodominant epitope of ConM, we tested whether the autologous ConM NAb responses in rabbit sera could be blocked by pre-incubation with either ConM SOSIP trimers or ConM SOSIP trimers in which the V1V2 region was swapped with that of BG505. Whereas the ConM SOSIP trimer completely blocked the autologous ConM NAb responses, the ConM SOSIP trimer with the BG505 V1V2 region had almost no effect in the majority of animals, confirming that the trimer apex is critical for ConM NAb responses (Fig. 6d). To support this finding, competition ELISA was performed with the apex-proximate bNAbs PGT128, CH01, PG16, gl-PG9, and VRC01. Consistent with a V1V2-directed response, we observed strong competition of PGT128, CH01, PG16, and gl-PG9 with the sera from all groups that, in the case of PGT128, CH01, PG16, and gl-PG9, correlated significantly with ConM neutralization midpoint titers (Fig. 6e and Supplementary Fig. 7c). Epitope mapping of ConS neutralization showed a strong dependency on amino acid residues at position 173 and 324 located in the V2 loop and at the V3-base, respectively (Supplementary Table 4). The neutralization-reducing effect of the G324A mutation was significantly stronger with sera from the I53-50NP group than in the corresponding trimer group (Supplementary Fig. 7d).

Thus, competition ELISAs, neutralization-blocking, and relative neutralization of mutant viruses, all supported the hypothesis that the location of the immunodominant neutralizing epitopes on SOSIP trimers determine whether they benefit from presentation at high density on I53-50NPs.

## Discussion

In this study, we describe the computational design, production, in vitro and in vivo characterization of a self-assembling two-component NP system presenting native-like Env trimers. We used several techniques to show that we have produced mono-disperse, icosahedral NPs that assemble in vitro and present 20 native-like SOSIP trimers in a well-ordered geometrical structure. The remarkable efficiency with which the various clades of NPs assembled and the intact structure of the icosahedral I53-50NP core in the presence of the displayed SOSIP trimers are testaments to the robustness and versatility of the computationally designed I53-50NP.

The SOSIP-I53-50NPs have several interesting physical features that may contribute to their potential to enhance SOSIP immunogenicity. First, the overall size of the NP immunogens is in accordance with that suggested to be optimal for uptake into the lymphatic system (40–50 nm)[44–46]. Second, the SPR experiments showed that the spacing between SOSIP trimers on the I53-50NPs enables bivalent interactions between certain mAbs and their epitopes (presumably on neighboring SOSIP trimers), providing another potential layer of avidity to BCR binding in addition to particulate display. Third, although not exploited in this study, the fact that these particles assemble in vitro and have a hollow interior may allow controllable encapsulation of adjuvants or TRL agonists. Indeed, recent studies have shown that proteins and RNA can be efficiently encapsulated into the I53-50NP lumen[30,47].

The two-component SOSIP-I53-50NP system offers several considerable advantages over previously described Env-presenting NPs such as enveloped VLPs and ferritin NPs[17,18,20,22,25]. In contrast to VLPs, SOSIP-I53-50NPs are produced in good yield and can be scaled up using standard methods for biologics production in a good manufacturing practice setting. In contrast to enveloped VLPs and ferritin NPs, SOSIP components are efficiently cleaved and can be purified by PGT145-affinity chromatography prior to particle assembly, ensuring that only native-like trimers are presented on the NP. The intrinsic inability of SOSIP-ferritin NPs to present a homogeneous array of native-like fully cleaved trimers is likely to explain the strong V3-mediated non-NAb responses (Tier-1 neutralization) observed in ferritin immunization studies[18,22,48]. As high-affinity non-neutralizing B-cell lineages such as those targeting the V3 loop may outcompete neutralizing B-cell lineages, it is advantageous to have a NP platform that minimizes the elicitation of non-neutralizing Abs[43]. Indeed, compared with the homomeric ferritin immunogen, the two-component SOSIP-I53-50NPs improved the quality of Env-specific Ab responses as illustrated by the significantly lower neutralization of the Tier-1 viruses SF162 and MW965.25.

With ~40- and 10-fold higher median autologous neutralization after a single immunization than ConM SOSIP and ConM SOSIP-ferritin, respectively, ConM SOSIP-I53-50NP was particularly effective at priming the immune response. However, this difference declined after multiple immunizations. One potential explanation for this phenomenon is that after priming, antigen-specific B cells may have already undergone enough affinity maturation that the higher avidity provided by multivalent presentation is no longer necessary during boosting. This hypothesis is consistent with a recent study clearly demonstrating that the benefits of multivalent antigen presentation are most evident in the context of low-affinity antigen-BCR interactions[49]. Alternatively, or concurrently, the induction of I53-50NP-specific antibodies may have a negative effect on the boosting potential of SOSIP-specific Ab responses, although our data showed that SOSIP-targeting Ab responses were boosted much more strongly than anti-I53-50NP responses. In either case, priming with NPs and boosting with trimers may further improve Env immunogenicity beyond the enhancements reported here. Additional immunization studies will be necessary to determine how SOSIP-I53-50NPs can be optimally employed.

Epitope mapping of the antibody responses to BG505 SOSIP suggested reduced targeting of the gp120-gp41 interface epitopes when SOSIP is presented on I53-50NPs. Because of its immunodominant epitope located close to this region, the BG505 SOSIP trimer presented on NPs elicited restricted responses. Along with the lower immunogenicity of interface epitopes, we observed a significantly lower immunogenicity of the highly dominant and non-neutralizing trimer base epitope[42,43]. As hiding the trimer base may enhance activation and affinity maturation of NAb-lineages, the ability to decrease base-targeting non-NAb responses is a promising feature of SOSIP-I53-50NPs. In addition, the ability of SOSIP-I53-50NPs to shape the vaccine-induced response by reducing Ab responses to the gp120-gp41 interface may be of considerable value for certain germline-targeting immunization strategies where the aim is to focus Ab responses to a specific apex-proximal epitope[26,50].

Taken together, the results from the immunization studies with BG505 and ConM immunogens and the epitope mapping performed here suggest that the I53-50NP appears to be ideal for displaying SOSIP trimers that harbor an immunodominant epitope near the trimer apex. The key concept our data highlight is that the detailed geometry of epitope presentation—the structural context of the epitope with respect to neighboring antigen molecules and the nanoparticle core—matters. For a given immunogen, obtaining maximal immune responses may require optimizing the geometry of epitope presentation. Ideally, knowledge of the most potent neutralizing epitope locations on the antigen would inform this optimization. In this way, the ability to accurately design new NP scaffolds with custom structures (e.g., shapes, sizes, and symmetries) using Rosetta[30,33,51,52] may open up new avenues to the structure-based design of potent NP immunogens.

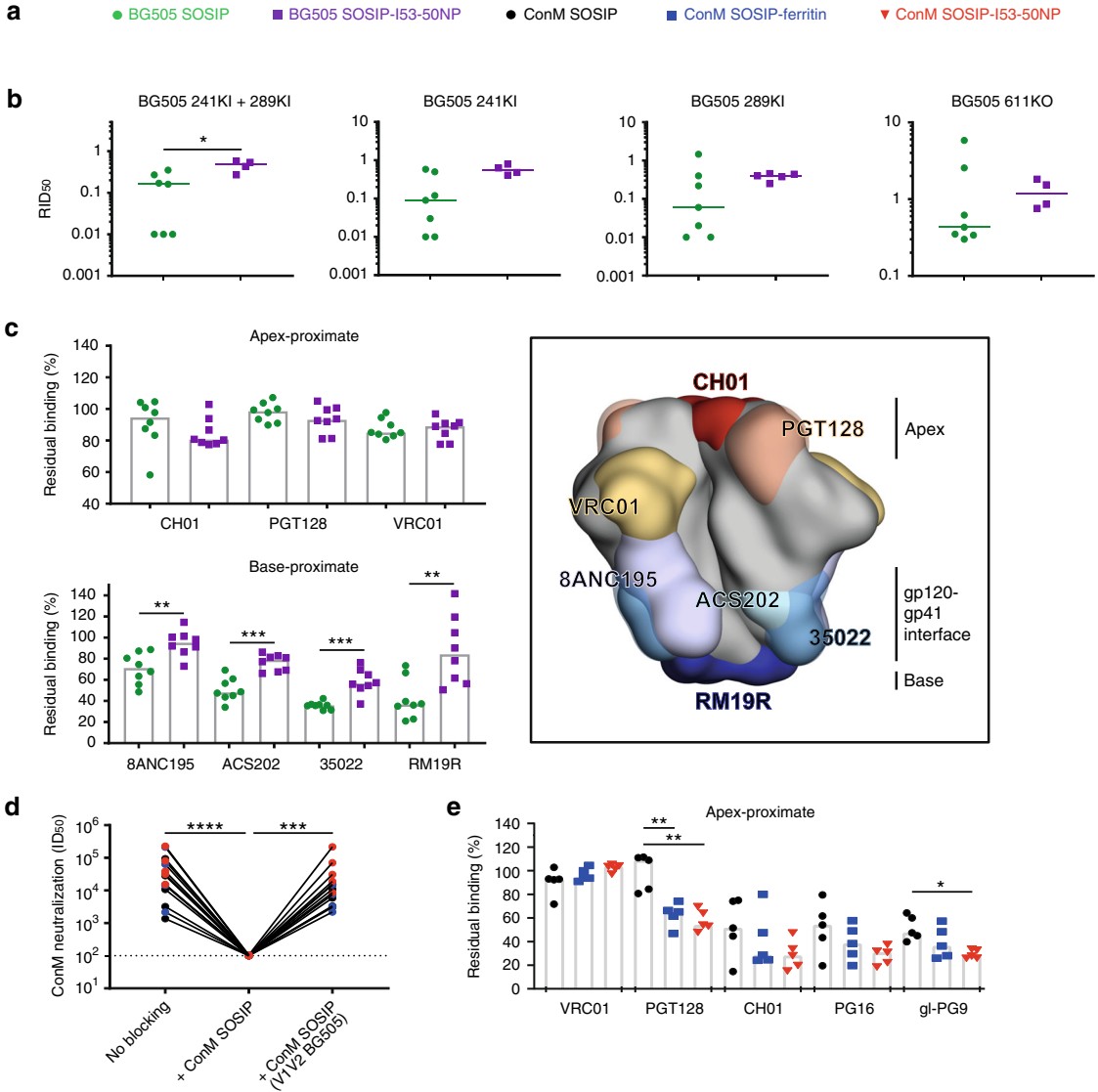

**Fig. 6** Epitope mapping of BG505 and ConM SOSIP-I53-50NP-induced (N)Ab responses. **a** Color coding for **b**–**e**. **b**–**e** Statistical differences between two groups were determined using unpaired two-tailed Mann–Whitney $U$-tests (*$p < 0.05$; **$p < 0.01$; ***$p < 0.001$). **b** Ratio of midpoint neutralization titers of several BG505 virus mutants relative to the parental BG505 virus is plotted (RID$_{50}$). Mutants shown are BG505 with glycans knocked in at position 241 and 289 (BG505 241KI+289KI), position 241 (BG505 241KI), position 289 (BG505 289KI), or a glycan knocked out at position 611 (BG505 611KO). Epitope mapping was only performed for sera from rabbits that had an ID$_{50}$ > 100 ($n = 7$ individual rabbits, for BG505 SOSIP recipients; $n = 4$ individual rabbits for BG505 SOSIP-I53-50NP recipients). See also Supplementary Table 3. Horizontal bars indicate the median. **c** Residual binding of bNAbs CH01, PGT128, VRC01, 8ANC195, ACS202, 35O22, and non-NAb RM19R after incubation of sera ($n = 8$ individual rabbits) with immobilized BG505 SOSIP in ELISA (left) and surface model of BG505 SOSIP (gray) with the corresponding Ab footprints highlighted. Bars indicate the median. **d** Midpoint autologous ConM neutralization in the absence of a blocking agent (no blocking), with ConM SOSIP added (+ConM SOSIP) or with ConM SOSIP that has the V1V2-loop of BG505 (+ConM SOSIP (V1V2 BG505)). Statistical differences between the three groups ($n = 5$ individual rabbits) was determined using a Friedman test followed by a Dunn's multiple comparisons test (***$p < 0.001$; ****$p < 0.0001$). **e** Residual binding of bNAbs VRC01, PGT128, CH01, PG16, and gl-PG9 after incubation of sera with immobilized ConM SOSIP in ELISA. Bars indicate the median ($n = 5$ individual rabbits). Source data are provided as a Source Data file

In conclusion, we have developed a NP platform that has the potential to enhance and shape the induction of antibodies targeting neutralizing epitopes, while decreasing the stimulation of non-neutralizing lineages, through the presentation of homogeneous arrays of native-like HIV-1 Env trimers. SOSIP-I53-50NPs are a promising priming immunogen for immunization strategies aiming to increase and broaden neutralizing antibody responses against HIV-1.

## Methods

**Computational design of SOSIP-I53-50A**. A recently described docking protocol, sic_axle[35] was used to dock the crystal structure of BG505 SOSIP.664 (PDB ID 4TVP)[34] to various scaffolds with exterior-facing termini from a set of self-assembling protein nanomaterials. The protocol was based on the previously described two-component docking protocol[33]. The protocol minimizes the distance between the two trimeric proteins and optionally the distance between user-specified termini, while preventing clashes, defined as interatomic distances below a user-defined distance threshold. The degrees of freedom searched during docking

are the displacement along the shared symmetry axis of one of the two proteins (r) and its rotation about the threefold symmetry axis (ω). In this case, the docking protocol minimized the distance between the C termini of the SOSIP trimer and the N termini of each NP trimer. Docking results were inspected manually and flexible genetic linkers were designed for the resulting fusion proteins.

**Construct design**. To create the BG505 SOSIP.v5.2-I53-50A construct, a modified I53-50A sequence (see below) was ordered (Integrated DNA Technologies) and cloned by Gibson assembly into a BamHI–NotI-digested pPPI4 plasmid containing the previously described BG505 SOSIP.v5.2 sequence[30,36]. The BamHI site (amino acids GS) is located between the final codon of the SOSIP trimer component (that for residue-664) and the first I53-50A codon. The modification to the I53-50A sequence comprised the replacement of the first two amino acids (MK) with a GGSGGSGGSGGSEKAAKAEEAARK linker. This linker comprises two sections: a flexible N-terminal section (GGSGGSGGSGGS) and a helical C-terminal section (EKAAKAEEAARK). The C-terminal section serves the purpose of extending the N-terminal helix of the I53-50A subunit towards the exterior surface of the nanoparticle. The flexible GGS linker was chosen to warrant the independent folding of the two components that make up SOSIP.v5.2-I53-50A and covers the distance between the termini of the docked SOSIP and I53-50A trimers (including the I53-50A helical extension). In our docked model this distance was 16 Å. Considering the heuristic of 1–2 amino acids per 2 Å linear distance, a 12-residue linker was chosen. The same cloning procedure and linker were used to produce the I53-50A.1NT1, I53-50A.1NT2, and I53-50A.1PT1 variants.

The SOSIP-I53-50A constructs based on other Env genotypes (AMC011, ZM197M, and ConM) were produced by replacing the SOSIP trimer component of the BG505 SOSIP.v5.2-I53-50A plasmid with the relevant Env sequence, using PstI–BamHI digestion followed by ligation or Gibson assembly. The ConM SOSIP.v7 and ConM SOSIP.v7-ferritin sequences have been described elsewhere[22]. The ZM197M SOSIP.v5.2(519S, 568D, 570H, 585H) construct was produced by replacing the ZM197M SOSIP.v5.2 sequence that has been described elsewhere[36] by Gibson assembly with a gBlock Gene fragment (Integrated DNA Technologies) of ZM197M SOSIP.v5.2 sequence that has the previously described I519S, L568D, V570H, and R585H substitutions introduced[26]. To generate the AMC011 SOSIP. v8.1 sequence, the previously described AMC011 SOSIP.v4.2 plasmid[37] was replaced by Gibson assembly with a gBlock Gene fragment (Integrated DNA Technologies) of AMC011 SOSIP.v4.2, which contained the following mutations: the v5.2 substitutions A73C and A561C[36]; the trimer-stabilizing substitutions, E47D, T49E, V65K, E106T, M165L, E429R, K432Q, K500R[39]; and four additional gp41-stabilizing changes, F519S, L568D, V570H, R585H[26]. For sequences of these constructs, see Supplementary Methods.

For Ni-NTA ELISAs, competition ELISAs and SPR experiments, His-tagged versions of the corresponding SOSIP trimer constructs without the I53-50A extension were used. Each was generated by addition of a GSGSGGGSGHHH HHHHH amino acid sequence immediately after the C-terminal residue-664 of the trimer. For competition ELISAs, D7324-tagged SOSIP trimers were also used, which were generated by adding the sequence GSAPTKAKRRVVQREKR after residue-664. The underlined GS sequence contains a BamHI site.

For neutralization depletion experiments, various ConM and BG505 SOSIP variants were produced. All constructs had the D368R mutation introduced. The ConM SOSIP.v7 (V1V2 BG505) sequence was produced by cloning in a ConM SOSIP.v7 gene fragment that contained the V1V2 region of BG505 (residue 131–196, HxB2 numbering) in the ConM SOSIP.v7 plasmid by Gibson assembly[22]. To make the BG505 SOSIP construct that had the 241 and 289 glycan knocked in, the following mutations were introduced into a stabilized BG505 SOSIPv5.2: P240T, S241N, M271I, F288L, T290E, and P291S.

**Env protein expression and purification**. All Env constructs described above were expressed in transiently transfected HEK293F cells (Invitrogen, catalog number R79009) essentially as reported previously[7,22], but with the following modifications for SOSIP-I53-50A constructs and variants. Specifically, to ensure optimal furin-mediated cleavage of the Env component, the cells were transfected with 250 µg of the SOSIP-I53-50A plasmid and 83.3 µg of a furin plasmid (an increase from the standard 62.5 µg amount). All SOSIP fusion proteins and ConM SOSIP.v7-ferritin were purified from vacuum-filtered (0.2 µm filters) transfection supernatants by PGT145 bNAb-affinity chromatography as described for ConM SOSIP-ferritin NPs elsewhere[22]. Protein concentrations were determined by the Nanodrop method using the proteins peptidic molecular weight and extinction coefficient. These values were obtained by filling in the proteins amino acid sequence in the online Expasy software (ProtParam tool).

**I53-50B.4PT1 and I53-50A.1NT1 protein purification**. Bare I53-50A.1NT1 component (without Env genetic fusion) was produced in the following manner. First, a pET29b+ plasmid containing the I53-50A.1NT1 gene was transformed into T7 Express Escherichia coli cells (New England Biolabs, catalogue number C2566). Terrific Broth media with 50 µg mL⁻¹ kanamycin was inoculated with a resulting colony and incubated overnight at 37 °C with 250 r.p.m. shaking to produce a starter culture. Two 2 L baffled flasks each containing 1 L of Luria-

Bertani media with 50 µg mL⁻¹ kanamycin were inoculated with 25 mL of starter culture and incubated with 200 r.p.m. shaking at 37 °C until OD600 reached ~0.6–0.8, at which point isopropyl β-D-1-thiogalactopyranoside was added to 1 mM. We continued to incubate the culture for another 3 h under the same conditions. Cells were collected by centrifugation at $4000 \times g$ for 10 min and media decanted. Cell pellets were resuspended in 25 mL lysis buffer (50 mM Tris pH 8.0, 0.5 M NaCl, 20 mM imidazole, 1 mM dithiothreitol (DTT), 1 mM phenylmethyl sulfonyl fluoride, 0.05 mg mL⁻¹ DNAse, 0.05 mg mL⁻¹ RNAse, 0.1 mg mL⁻¹ lysozyme, 5% glycerol, 0.75% 3-[(3-cholamidopropyl)dimethylammonio]-1-pro-panylsulfonate (CHAPS)) and then sonicated. Lysate was clarified by centrifugation at $33,000 \times g$ for 20 min at 4 °C. Supernatant was applied to a 5 mL HisTrap HP column (GE Healthcare) equilibrated with 50 mM Tris pH 8.0, 500 mM NaCl, 20 mM imidazole, 1 mM DTT after filtering through a 0.22 µm filter. The column was washed with ~10 CV of equilibration buffer, then eluted over a gradient to 500 mM imidazole in equilibration buffer. Fractions containing I53-50A.1NT1 were concentrated in a Vivaspin filter with a 10 kDa molecular weight cutoff (GE Healthcare) before further purification on a Superdex200 Increase 10/300 GL SEC column (GE Healthcare) using 25 mM Tris pH 8, 500 mM NaCl, 0.75% CHAPS, 1 mM DTT buffer.

To produce I53-50B.4PT1, a pET29b+ plasmid containing the I53-50B.4PT1 gene was transformed into T7 Express E. coli cells (New England Biolabs, catalog number C2566). The following steps were essentially as described for I53-50A.1NT1 above. The only modification is the supplementation of 0.75% CHAPS to the equilibration buffer for the HisTrap HP column.

In order to remove endotoxin, purified I53-50B.4PT1 was immobilized on Ni2+-NTA resin in a 5 mL HisTrap HP column (GE Healthcare) equilibrated with the following buffer: 25 mM Tris pH 8, 500 mM NaCl, 0.75% CHAPS. Immobilized I53-50B.4PT1 was then washed with ~10 CV of the equilibration buffer. The protein was eluted over gradient to 500 mM imidazole in equilibration buffer. Fractions containing I53-50B.4PT1, which elutes around ~175 mM imidazole, were concentrated in a Vivaspin filter with a 10 kDa molecular weight cutoff and subsequently dialyzed twice against equilibration buffer (GE Healthcare).

**I53-50NP assembly**. Concentrations of purified individual nanoparticle components were determined by measuring absorbance at 280 nm using a UV-Vis spectrophotometer (Cary 8454, Agilent Technologies) and extinction coefficients (obtained by inserting the sequence in the online Expasy software (ProtParam tool). The following assembly steps were performed on ice: I53-50A.1NT1 was added to an eppendorf tube to a final concentration of 50 µM in the in vitro assembly reaction. Assembly buffer I (25 mM Tris pH 8, 500 mM NaCl, 0.75% CHAPS) was then added to a volume of 1 mL. Finally, I53-50B.4PT1 pentamer was added to the tube for a final concentration of 50 µM. Assemblies were incubated at room temperature with gentle rocking for at least 1 h before subsequent purification by SEC using a Superose 6 Increase 10/300 GL column (GE Healthcare). Assembled particles elute at around 11.5 mL from this column. Assembled NPs were centrifuged for 10 min at $21,000 \times g$ and 4 °C or sterile filtered (0.22 µm) immediately before column application.

**SOSIP-I53-50NP assembly**. After PGT145-purification (see Env protein expression and purification), SOSIP fusion proteins were passed through a Superose 6 Increase (GE Healthcare) SEC column in Assembly Buffer II (25 mM Tris, 500 mM NaCl, 5% glycerol pH 8.2) to remove aggregated proteins. The glycerol component was included in the Assembly Buffer II to minimize aggregation of SOSIP fusion proteins during the assembly of NPs, but we found that their presence increased the recovery of the assembled NPs during the concentration and dialysis stages described below. Initial BG505 SOSIP-I53-50A assemblies also had 250 mM L-Arginine in the Assembly buffer II in an attempt to reduce aggregation. However, as it later turned out that this supplement did not decrease aggregation, it was left out in later assemblies. After the SEC procedure, the column fractions containing non-aggregated SOSIP-I53-50A(.1NT1) proteins were immediately pooled and mixed in an equimolar ratio with I53-50B.4PT1 (produced as described above) for an overnight (~16 h) incubation at 4 °C. The assembly mix was then concentrated at $350 \times g$ using Vivaspin filters with a 10 kDa molecular weight cutoff and passed through a Superose 6 Increase column in Assembly Buffer II (GE Healthcare). The fractions corresponding to the assembled NPs (elution between 8.5 and 10.5 mL with a peak at 9 mL) were pooled and concentrated at $350 \times g$ using Vivaspin filters with a 10 kDa molecular weight cutoff (GE Healthcare). NPs were then buffer exchanged into phosphate-buffered saline (PBS) by dialysis at 2 °C overnight, followed by a second dialysis step for a minimum of 4 h, using a Slide-A-Lyzer MINI dialysis device (20 kDa molecular weight cutoff; ThermoFisher Scientific). Nanoparticle concentrations were determined by the Nanodrop method using the particles peptidic molecular weight and extinction coefficient. To get these values, first the molecular weight and extinction coefficient of the SOSIP-I53-50A(.1NT1) and I53-50B.4PT1 components were obtained by filling in their amino acid sequence in the online Expasy software (ProtParam tool). The peptidic mass or extinction coefficient of SOSIP-I53-50NP was then calculated by summing the obtained peptidic masses or extinction coefficient, respectively, of each component of the NP.

**SDS-PAGE analysis**. Two micrograms of SOSIP trimer or 3.2 μg of SOSIP-I53-50NP (which is equivalent to 2 μg of trimer) were loaded on a 4–12% Tris-Glycine gel or 6–18% Tris-Glycine gel, respectively (both from Invitrogen). SDS-PAGE (polyacrylamide gel electrophoresis) was performed as described previously[7].

**Blue native PAGE analysis**. Blue native PAGE analysis was performed as described elsewhere[7]. Two micrograms of SOSIP trimer or 3.2 μg of SOSIP-I53-50NP were mixed with loading dye and run on a 4–12% Bis-Tris NuPAGE gel or 3–12% Bis-Tris NuPAGE gel (both from Invitrogen), respectively.

**Ni-NTA-capture ELISA**. A 1 μg mL$^{-1}$ concentration of His-tagged SOSIP protein in Tris-buffered saline (TBS) was used to coat wells of a 96-well Ni-NTA ELISA plate for 2 h (Qiagen). After three washes with TBS, the wells were blocked for 30 min with casein/TBS (ThermoFisher Scientific). A 1 μg mL$^{-1}$ concentration of antibody in casein/TBS was serially diluted 1:3 and incubated at room temperature (RT). Plates were washed after 2 h with TBS and a 1:3000 dilution of goat anti-human horseradish peroxidase (HRP)-conjugated antibody (109-035-008; Jackson Immunoresearch) in casein/TBS was added for 45 min before binding was quantified. Next, after washing the plates five times with TBS + 0.05% Tween-20, developing solution (1% 3,3′,5,5′-tetranethylbenzidine (Sigma-Aldrich), 0.01% H$_2$O$_2$, 100 mM sodium acetate, and 100 mM citric acid) was added. Development of the colorimetric endpoint proceeded for 3 min before termination by adding 0.8 M H$_2$SO$_4$.

**Surface plasmon resonance**. Antibody binding to soluble SOSIP trimers or SOSIP-I53-50NPs was analyzed by SPR. All SPR analyses were conducted on a BIAcore 3000 instrument at 25 °C, with HBS-EP (10 mM HEPES, 150 mM NaCl, 3 mM EDTA, 0.005% surfactant P20) as running buffer. His-tagged trimers and NPs were captured onto C1 sensor chips by an anti-His antibody that had been covalently coupled to the chips (as previously described[53]).

We analyzed IgG and Fab binding to SOSIP trimers and SOSIP-I53-50NPs, which were immobilized at low levels, to minimize mass-transport limitations, but with the same amounts of Env for trimers alone and on NPs ($R_L$ values attributable to Env of 160 ± 1.6 RU for BG505 and 170 ± 0.34 for ConM). IgGs were injected at the concentrations indicated in the figures and the flow rate was 30 μL min$^{-1}$. For Fabs, the flow rate was set to the maximum, 50 μL min$^{-1}$, in order to minimize mass-transport limitation. Absence of mass-transoprt limitation was documented as described in Supplementary Methods. Each Fab was titrated from the highest concentration used (either 2 or 1 μM) in twofold dilutions until the signal was undetectable. The binding data thus obtained were analyzed by fitting a Langmuir or, when required for sufficiently good fit, a conformational-change model, as included in the Biaevaluation software (GE).

Conversely, we also analyzed the binding of SOSIP trimers and SOSIP-I53-50NPs to immobilized Abs (see Supplementary Methods). Initially, binding to a panel of 13 NAbs and non-NAbs was analyzed for SOSIP.v5.2 trimers at 40 nM and SOSIP-I53-50NPs at 2 nM, i.e., with the same amount of Env per volume. The analytes were next titrated at the same ratio against six selected bNAbs.

**N-glycan analysis by HILIC-UPLC**. N-glycans were enzymatically released from gel bands by digestion with Peptide-N-Glycosidase F (PNGase F, New England Biolabs) at 37 °C for 16 h. An aliquot of glycans was resuspended in 30 μl of water followed by the addition of 80 μl of labeling mixture (30 mg mL$^{-1}$ 2-AA and 45 mg mL$^{-1}$ sodium cyanoborohydride in a solution of sodium acetate trihydrate [4% w v$^{-1}$] and boric acid [2% w v$^{-1}$] in methanol). Samples were incubated at 80 °C for 1 h. Excess label and PNGase F was removed using Spe-ed Amide 2 cartridges (Applied Separations). Glycans were analyzed on a Waters Acquity H-Class UPLC instrument with a Glycan BEH Amide column (2.1 × 100 mm, 1.7 μM, Waters) and the following gradient: time ($t$) = 0: 22% A, 78% B (flow rate = 0.5 mL min$^{-1}$); $t$ = 38.5: 44.1% A, 55.9% B (0.5 mL min$^{-1}$); $t$ = 39.5: 100% A, 0% B (0.25 mL min$^{-1}$); $t$ = 44.5: 100% A, 0% B (0.25 mLmin$^{-1}$); $t$ = 46.5: 22% A, 78% B (0.5 mL min$^{-1}$), where solvent A was 50 mM ammonium formate (pH 4.4) and B was acetonitrile. Fluorescence was measured at an excitation wavelength of 250 nm and an emission wavelength of 428 nm. Data were processed using Empower 3 software. The relative abundance of oligomannose-type glycans were determined by digesting fluorescently labeled glycans with endoglycosidase H at 37 °C for 16 h. Glycans were extracted using a polyvinylidene fluoride protein-binding membrane (Millipore) and analyzed as described above.

**Negative-stain EM**. Negative-stain EM experiments were performed as described previously[54,55]. SOSIP trimers and SOSIP-I53-50NPs samples were diluted to 20–50 μg mL$^{-1}$ and loaded onto the carbon-coated 400-mesh Cu grid that had previously been glow-discharged at 15 mA for 25 s. Grids were negatively stained with 2% (w v$^{-1}$) uranyl formate for 60 s. Data collection was performed on a Tecnai Spirit electron microscope operating at 120 keV. The magnification was ×52,000 with a pixel size of 2.05 Å at the specimen plane. The electron dose was set to 25 e$^{-}$ Å$^{-2}$. All imaging was performed with a defocus value of −1.50 μm. The micrographs were recorded on a Tietz 4 × 4k TemCam-F416 CMOS camera using Leginon automated imaging interface. Data processing was performed in Appion data processing suite. With nanoparticle samples, ~500–1000 particles were

manually picked from the micrographs and 2D-classified using the Iterative multivariate statistical analysis (MSA)/multireference alignment (MRA) algorithm. With trimer samples, 10,000–40,000 particles were auto-picked and 2D-classified using the Iterative MSA/MRA algorithm.

**Differential scanning fluorimetry**. All measurements were performed on a Prometheus NT.48 NanoDSF instrument (NanoTemper Technologies) as previously described[55]. Briefly, SOSIP trimer, SOSIP-I53-50A(.1NT1) protein, and SOSIP-I53-50NP samples were diluted to 0.5–1 mg mL$^{-1}$ and loaded into standard grade glass capillary tubes (NanoTemper Technologies). The temperature was increased linearly from 25 °C to 95 °C at a rate of 1 °C min$^{-1}$. The instrument software was applied for data processing and $T_m$ estimation.

**Cryo-EM data collection**. Three microliters of purified BG505 SOSIP-I53-50NP at ~1 mg mL$^{-1}$ was plunge-frozen manually on holey carbon grids (2 × 2 C-flatTM). Digital micrographs were collected on an FEI Titan Krios operating at 300 keV with a K2 Summit direct electron detector (Gatan, Inc.) controlled with Leginon automated microscopy software[56]. Each movie micrograph was composed of 32, 250 ms frames collected with a dose rate of ~10 e$^{-}$ pix$^{-1}$ s$^{-1}$ over a defocus range of −1.25 to −5.1 μm and at a magnification of 38,168 resulting in a final pixel size of 1.31 Å. Movie frames were aligned and dose-weighted with MotionCor2[57] and CTF parameters were calculated with GCTF[58].

**Cryo-EM data processing**. Individual projection images (3996) were selected manually from 1445 micrographs using custom software and exported as coordinate files into Relion/2.0 for all subsequent single-particle processing[59]. Initial particle picks were extracted from dose-weighted micrographs and binned four times. Two rounds of reference-free 2D classification were performed, followed by subset selection. The first round included the entire particle width, whereas the second round was carried out with a mask around just the I53-50NP components, resulting in classes significantly closer to uniqueness (Supplementary Fig. 1c). This clean set of particles was re-centered, re-extracted, and binned twice. An initial round of 3D autorefinement was carried out using the I1 symmetry class without any masking. To improve alignment, a soft mask in the shape of just the I53-50NP was created and used in subsequent rounds of 3D classification and refinement (Supplementary Fig. 1d). At this step, particles were re-centered and re-extracted one more time as unbinned particles. One or more rounds of 3D classification without alignment followed by subset selection were then performed, followed by additional rounds of 3D autorefinement until no additional gains in resolution were observed, resulting in a final set of 3590 particles. The 3D mask used for refinement was then used to sharpen the final map using Relion auto-B-factor of −149 and the map was multiplied by the microscope's modulation transfer function. The final resolution of 4.5 Å is reported as the resolution where the goldstandard Fourier shell correlation drops below 0.143 (Supplementary Fig. 1e).

**Cryo-EM data model building**. The computationally predicted model for the I53-50NP asymmetric unit was fitted into the sharpened cryo-EM map with UCSF Chimera[60]. The model was then idealized, trimmed of hydrogen, and refined into the map with icosahedral symmetry using RosettaRelax[61]. Three hundred and eighteen unique models were generated and evaluated for quality with EMRinger[62] and MolProbity[63], and the model with the best cumulative score was chosen. Finally, a single round of Phenix real-space refinement was performed with symmetry[64]. Whole model Cα root-mean-square deviations between predicted and relaxed models were calculated in UCSF Chimera[60].

**Dynamic light scattering**. DLS was used to assess the hydrodynamic radius (Rh) and polydispersity of the assembled SOSIP-I53-50NPs. The particles were diluted to 0.025 μg mL$^{-1}$ in PBS and loaded into a Dynapro Nanostar instrument (Wyatt Technology Corporation). Rh and polydispersity values were measured with ten acquisitions of 5 s each at 25 °C and analyzed using the manufacturer's software (Dynamics, Wyatt Technology Corporation), while assuming particles with a spherical shape.

**B-cell activation assay**. WEHI-231 B-cell lines expressing the specific bNAbs (VRC01, PGT145, and PGT121) were donated by Takayuki Ota and David Nemazee at The Scripps Research Institute and were maintained in advanced Dulbecco's Modified Eagle's Medium (DMEM) supplemented with 10% fetal calf serum, β-mercapto-ethanol (55 μM), L-glutaMAX (2 mM), penicillin (100 U mL$^{-1}$), and streptomycin (100 μg mL$^{-1}$) (advanced DMEM++). B-cell activation experiments were performed as described elsewhere[22,65]. Briefly, 1 day prior to the assay, cells were incubated with 1 μg mL$^{-1}$ doxycycline to induce the expression of BCRs. The next day, cells were suspended at 4 million cells per mL in advanced DMEM++, labeled with 1.5 μM Indo-1 (Invitrogen) for 30 min at 37 °C and washed with Hank's Balance Salt Solution containing 2 mM CaCl$_2$, followed by another incubation of 30 min at 37 °C. Aliquots of 1 million cells per mL were then stimulated at room temperature with SOSIP trimers, NPs, or control reagents. Doxycycline-induced WEHI-231 B cells were incubated with 5 μg mL$^{-1}$ of SOSIP trimers or the equimolar amount presented on SOSIP-ferritin or SOSIP-I53-50NPs.

Calcium ($Ca^{2+}$) signals were recorded on a LSR Fortessa (BD Biosciences) by measuring for 210 s the 405 per 485 nm emission ratio of Indo-1 fluorescence upon UV excitation. Kinetic analyses were performed using FlowJo v10.

**Rabbit immunizations.** For all immunization studies described here, female and naive New Zealand White rabbits of 2.5–3 kg, from multiple litters, were arbitrarily distributed among groups. Rabbits were sourced and housed at Covance Research Products, Inc. (Denver, PA, USA) and immunization were performed under permits with approval number C0135-016 and C0171-017. All immunization procedures complied with all relevant ethical regulations and protocols of the Covance Institutional Animal Care and Use Committee. For ConM immunogens, groups of five female New Zealand White Rabbits were given two intramuscular immunizations in each quadriceps at weeks 0, 4, and 20. The immunization mixture involved 30 µg of SOSIP trimers or an equimolar amount presented as ferritin-NPs (38 µg) or as SOSIP-I53-50NPs (48 µg) formulated in GLA-LSQ adjuvant (25 µg GLA and 10 µg QS21 per dose, IDRI). Calculations of the dose were based on the peptidic molecular weight of the proteins (thus disregarding glycans), which were obtained essentially as described above in Env protein expression and purification, and SOSIP-I53-50NP assembly. The rabbits were bled on the day of immunization and then at weeks 2, 6, 8, 12, 16, and 20. For BG505 immunogens, the same protocol was used, except that the rabbit group size was 8 and the schedule involved immunizations at weeks 0, 4, 8, and 20. Instead of a bleed at week 12, a bleed was taken at week 10.

**Serum antibody ELISA.** A 6.5 nM concentration of His-tagged versions of the BG505 or ConM SOSIP trimer immunogens in TBS was added to a 96-well Ni-NTA plate (Qiagen) for 2 h at RT. The plates were blocked for 30 min with TBS + 2% skimmed milk. Threefold serial dilutions of BG505-immunized rabbit sera, starting from a minimum of 1:200 dilution, were added in TBS + 2% skimmed milk + 20% sheep serum for a 2 h incubation at RT. For ConM-immunized rabbit sera a starting dilution of 1:500 was used. A 1:3000 dilution of HRP-labeled goat anti-rabbit IgG (111-035-144; Jackson Immunoresearch) in TBS + 2% skimmed milk was added for 1 h at RT. Up to this point, in between each step, the plates were washed three times with TBS. Next, after washing the plates five times with TBS + 0.05% Tween-20, developing solution (1% 3,3′,5,5′-tetranethylbenzidine (Sigma-Aldrich), 0.01% $H_2O_2$, 100 mM sodium acetate, and 100 mM citric acid) was added. Development of the colorimetric endpoint proceeded for 3 min before termination by adding 0.8 M $H_2SO_4$. The same procedure was used to measure binding Ab titers against the unmodified I53-50NP, except that the I53-50NP coating concentration was 3 nM and the serum starting dilution was a minimum of 1:1000.

**Competition ELISA.** The assay was performed essentially as described elsewhere[22] and general procedures (i.e., wash steps, blocking, and colorimetric detection) are described in Serum antibody ELISA above. Briefly, Ni-NTA (Qiagen) or D7234-coated plates (half-area plate coated overnight with Ab D7324 (Aalto Bioreagents) at 10 µg mL$^{-1}$ in 0.1 M NaHCO$_3$ pH 8.6) were loaded for 2 h at RT with 6.5 nM of His- or D7324-tagged ConM SOSIP or BG505 SOSIP trimers. Subsequently, serum was added to a final dilution of 1:100 for 30 min after which mAbs were added for 1.5 h at a concentration that gave 80% of the maximal binding signal as assayed in a previous titration experiment in the absence of serum (0.1 µg mL$^{-1}$: PG16; 0.2 µg mL$^{-1}$: VRC01, PGT128; 1 µg mL$^{-1}$: gl-PG9; 2.5 µg mL$^{-1}$: CH01; 4 µg mL$^{-1}$: 35O22; 5 µg mL$^{-1}$: 8ANC195, ACS202; RM19R). Binding of mAbs was detected by 1:3000 dilution of HRP-labeled donkey anti-human IgG conjugate (709-035-149; Jackson Immunoresearch). Residual binding was determined by:

$$\%\text{Residual binding} = \frac{(OD_{450} - OD_{450}\text{neg})}{OD_{450}\text{pos}} \cdot 100 \qquad (1)$$

where $OD_{450}$neg is the signal of the wells where no mAb was added (negative control) and $OD_{450}$pos the signal of the wells where no serum was added (positive control).

**Neutralization assay.** Neutralization assays based on the use of Env-pseudotyped viruses and TZM-bl cells were performed as described elsewhere[6,66]. In general, sera were diluted to 1:20 and then serially by threefold before mixing with Env-pseudotyped virus. However, for sera from ConM SOSIP-I53-50NP-immunized rabbits taken at weeks 6, 8, 12, 16, 20, and 22, the initial dilution was 1:100. Neutralization assays were performed at Amsterdam University Medical Center in Amsterdam, The Netherlands (as described in ref. [6]) and at Duke University Medical Center in Durham, NC, USA (as described in ref. [66]). Midpoint neutralization titers (ID$_{50}$-values) were determined as the serum dilution at which infectivity was inhibited by 50%. The neutralization depletion experiments were performed as described previously[22]. Briefly, rabbit sera were incubated for 1 h with 40 µg mL$^{-1}$ of ConM SOSIP.v7 or ConM SOSIP.v7 (V1V2 BG505), or in the case of BG505 immunogen-recipients BG505 SOSIP or BG505 SOSIP 241 + 289KI. The following steps were the same as a regular neutralization assay (described above). All depletion constructs were PGT145-purified and contained a D368R mutation to abrogate binding to CD4 on the TZM-bl cells.

**Statistical analysis.** Comparisons between two rabbit groups were performed with an unpaired two-tailed Mann–Whitney test. In one instance, where three paired groups were compared, a Fieldman test with Dunn's post test was used. Correlations were analyzed by calculating Spearman's rank correlation coefficients. Graphpad Prism 7.0 was used for statistical analyses.

**Reporting summary.** Further information on research design is available in the Nature Research Reporting Summary linked to this article.

## Data availability

A reconstructed density map and refined coordinates of the I53-50NP core of BG505 SOSIP-I53-50NP have been deposited in the Electron Microscopy Database (EMD-20261) and the Protein Databank (6P6F). All other data supporting the findings in this manuscript are available from the corresponding authors (R.W.S. and N.P.K.) upon reasonable request. The source data underlying Figs. 1–6 and Supplementary Figs. 1, 2, 3, 6 and 7 are provided as a Source Data file.

## Code availability

A static executable of the sic_axle program used to dock SOSIP to I53-50A is available upon request (neilking@uw.edu).

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

## Acknowledgements

We thank Diantha Verwey, Lauren Carter, and Rashmi Ravichandran for their help with protein purifications and characterizations; Edith Schermer for help with serum ELISAs; Takayuki Ota and David Nemazee for kindly sharing the PGT145, VRC01, and PGT121 B-cell lines; Michel Nussenzweig, James Robinson, Dennis Burton, Peter Kwong, Mark Connors, John Mascola, and William Olson for donating antibodies and reagents directly or through the NIH AIDS Research and Reference Reagent Program; Alba Torrents de la Peña for assistance in creating the model in Fig. 6c; Nicole van der Wel, Henk van Veen, and Daisy Picavet from the Electron Microscopy Center Amsterdam at the Amsterdam University Medical Center for support and assistance with visual screening of SOSIP-I53-50NP assemblies; Ian Wilson for input and discussions. This work was supported by the U.S. National Institutes of Health Grant P01 AI110657 (to J.P.M., A.B.W., P.J.K., and R.W.S.) and NIAID Contract #HHSN27201100016C (to D.C.M.); by the Bill and Melinda Gates Foundation through the Collaboration for AIDS Vaccine Discovery (CAVD), grants OPP1111923 and OPP1132237 (to D.B., N.K., J.P.M., and R.W.S.), and OPP1115782 (A.B.W. and M.Cr.); by the European Union's Horizon 2020 research and innovation program under grant agreement No. 681137 (R.W.S. and M. Cr.); and by the Fondation Dormeur, Vaduz (to R.W.S.). R.W.S. is a recipient of a Vici grant from the Netherlands Organization for Scientific Research (NWO). M.J.G. is a recipient of an AMC Fellowship and a Mathilde Krim Fellowship from the American Foundation for AIDS Research (amfAR) (109514-61-RKVA). M. Cam. is a recipient of a fellowship from the Consejo Nacional de Ciencia y Tecnología of Mexico (CONACYT). The electron microscopy data were collected at Electron Microscopy Facility of the Scripps Research Institute. The Amsterdam Cohort Studies (ACS) on HIV infection and AIDS, a collaboration between the Amsterdam Health Service, the Academic Medical Center of the University of Amsterdam, Sanquin Blood Supply Foundation, and the Jan van Goyen Clinic, are part of The Netherlands HIV Monitoring Foundation and are financially supported by the Center for Infectious Disease Control of the Netherlands National Institute for Public Health and the Environment.

## Author Contributions

P.J.M.B., A.A., Z.B., J.P.M., A.B.W., P.J.K., N.P.K., and R.W.S. conceived and designed experiments. P.J.M.B., A.A., Z.B., A.Y., B.F., T.P.L.B., I.B., J.D.A., J.A.B., M. Cam., D.E., C.A.C., and A.J.B. performed the experiments. J.B.B., W.S., D.E., and N.P.K. performed computational docking and design. P.J.M.B., A.A., Z.B., A.Y., I.B., J.D.A., M. Cat., C.A.C., A.J.B., C.L., K.S., M. Cr., D.M., D.B., J.P.M., A.B.W., P.J.K., N.P.K., and R.W.S. analyzed and interpreted data. T.K. organized the rabbit immunization studies. M.J.G. provided unpublished mAbs. A.S., M. Cat., and I.M.S. provided unpublished SOSIP constructs. P.J.M.B., A.A., Z.B., B.F., J.D.A., P.J.K., N.P.K., and R.W.S. wrote the manuscript and J.A.B., M.J.G., K.S., L.J.S., M. Cr., D.B., J.P.M., and A.B.W. edited and/or provided input to the manuscript.

## Additional information

**Competing interests:** The authors declare no competing interests.

