## [Peer Review File · Nature Communications]

Reviewers' Comments:

Reviewer #1:

Remarks to the Author:

The manuscript addresses an important point in vaccine-design: the development of nanoparticles that present well-ordered multimeric recombinant protein immunogens. The design and experimental aspects of the project are exceptional. The successful production of self-assembling nanoparticles expressing a complex and notoriously unstable HIV envelope protein, is very impressive. The immunization studies are well controlled. I have no concerns with the interpretation of the results or with the conclusions made by the authors. Although the study is solid, the significance of its main immunological finding (enhanced development of autologous nAbs) is questionable. The goal of HIV envelope-based immunizations is to elicit tier 2, heterologous nAbs. Such nAbs were not developed in the present study, despite the differential exposure of epitopes at the base of SOSIP molecules expressed on the nanoparticles and the altered immunogenicity of that protein when expressed on nanoparticles versus in a soluble form. Since the first immunogenicity study with soluble BG505 SOSIP, many groups have reported that by optimizing the design of the immunogen, or by multimerizing it, an improvement in the development of autologous nAbs is achieved. Yet, heterologous nAbs have not been generated, as is the case here. It is therefore uncertain whether such immunogen-optimization efforts are necessary for the development of tier 2, heterologous nAbs by SOSIP immunogens. It is also questionable if the development of autologous nAbs by SOSIP immunogens will lead (or is required for) to the development of heterologous nAbs. In fact, in the only case where heterologous nAbs were elicited by BG505 SOSIP (Nature 2017) the kinetics of heterologous nAbs were similar to those of the autologous nAbs.

Fig 4c. Why was there no VRC01 B cell activation by SOSIP trimers?

Fig 6. It is unclear whether the presentation of SOSIP on the I53-50NP rather than on ferritin particles alters the epitope specificity of the autologous neutralizing antibody responses. In other words, is there an advantage of using I53-50NP particles?

Line 307 is incomplete

Lines 325 and 328. Supplemental figure 6d does not present data discussed in this paragraph.

Reviewer #2:

Remarks to the Author:

Brouwer et al. reported a combination of SOSIP trimer and self-assembled protein nanoparticles to increase the B cell activation and to shape the immunogenicity towards HIV-1 Env apex-proximal epitopes while suppressing the non-NAb responses. In my opinion, the advantages and disadvantages of using SOSIP-nanoparticles as immunogens were made clear through this study: nanoparticles could withstand freeze-thaw cycle, native-like antigenicity was extensively characterized with bNAbs and SPR, various genotypes of SOSIP were tested, etc. are all merits of this study; but nanoparticles also introduced steric clashes (hence the limited accessibility to epitopes below CD4bs), most neutralization effects were shown towards ConM and ConM is only Tier-1 virus, and the nanoparticle core proteins elicited unnecessary immune responses. I must say the paper started as an insightful and exciting study but ended with a bit disappointment (because we can not say the responses against Tier-2 ConS virus was improved when compared to that of SOSIP alone). Overall, I support this publication, but I would like to raise several points that might require further input from the authors to enhance the potential impact of this manuscript:

Protein design rationale

SOSIP-I53-50 is composed of three parts: Env trimer, NP I53-50, and the linker between them. Since the design of NP I53-50 and the stabilization of SOSIP trimer have been elaborated elsewhere, it may

be worth explaining more about how to choose the linker.

Specifically, the author said the docking protocol minimized the distance between SOSIP and NP, but then chose a seamlessly arbitrary linker: GGSGGSGGSGGSEKAAKAEAAARK. What is the optimized linker length and how the current linker fit into that?

The flexibility of the linker seems to be the reason that SOSIP was poorly resolved in cryo-EM. Is there any rationale for preference of a flexible linker than a rigid one?

The reasons for using I53-50 as the scaffold seems vague (l. 100-102). For example, I could say I32-19, -06, -28 (among many designs previously reported by the author) might better resemble the diameter of HIV virion (~120 nm), and they all contain larger hollow interior that can be functionalized than then I53-50 does.

The Env spikes on the surface of HIV are sparsely distributed, unlike the dense packing on NP as revealed by the EM structure. I am not sure whether the denser alignment of Env would be good or bad for its immunogenicity, maybe the authors can provide some insights.

Immunogen design and characterization

L. 107: why were there four variants of I53-50A? What mutations were introduced? Were these mutations of I53-50 also used for ConM, AMC011, ZM197M constructs to increase the overall stability of NPs?

Why was ZM197M-NP construct discarded at later stages? Based on Fig. 3 and Fig. S3, ZM197M-NP assembled equally well and were stable enough. Similar strong binding by bNAbs to ZM197M was also observed in Fig. S4.

Comparing Fig. 4 to Fig. S4 could lead to some confusion: in Fig. 4 VRC01 or PGT122 bNAbs bound NP with higher capacity but in Fig. S4 VRC01 or PGT122 exhibited stronger interactions towards the parental trimer. I do not know whether it is because IgGs were immobilized in Fig. 4 while NPs were immobilized in Fig. S4, but I think the readers may feel confused as well.

Table 2 was appreciated, but I would suggest the authors also make a dot plot of the simulated KD to clearly show which is higher and which is lower, for easy comparison.

Another confusion might happen for the last four paragraphs of Results. To map the immunodominant epitope of BG505-NP and ConM-NP, respectively, the authors applied different assays. For BG505, the knock-in virus was used for neutralization; for ConM, a competition neutralization assay was designed by incubating sera with SOSIP while the virus was not changed. In competition ELISA, different apex-specific bNAbs were used (PGT128 and VRC01 for BG505 PG16, and gl-PG9 for ConM). It seems there is a lack of consistency for characterizing the two NP constructs. Maybe I am missing some information or background knowledge, but the readers may feel similarly misled too, for example, could the pre-incubating sera with BG505 SOSIP be applied to BG505 neutralization?

Nanoparticle recipes

Although the authors suggested that self-assembling proteins are better than VLPs or liposomes, the adjuvant used in the immunization was a liposome. I wonder whether the hydrophobic interactions in the liposome would interfere with NPs.

I can not see much from Fig. S6a about structural integrity as claimed, I can only see the morphology of particles is different from that of Fig. S3c.

Recent studies using trimer-conjugated liposomes (e.g. Tokatlian et al., 2019) also suggested this liposome formulation can trigger greater antibody responses and enhance humoral immunity. So I am not sure protein NP is better than liposome.

It seems L. 59 directly copy the third sentence in the abstract; I suggest the authors could introduce the difference between NP and VLP at this place for readers who are not familiar with different formulas.

Other issues

Fig. 1b said 'I53-50A1. NT1', the main text said 'I53-50A. 1NT1'

Fig. S3a, DSF plot, it seems the I53-50NP alone (black) did not have a melting peak at 82 degrees, or the melting temperature is larger than 100 degrees. If the latter case is true, does it mean fusion with SOSIP reduced the stability of NPs?

Fig. 3b legend said Protein Stabilizing Cocktail, while Fig. S3e said sucrose or PSC, so which one is the cryoprotectant?

L. 183, it is better to state where the His-tag located.

Reviewer #3:

Remarks to the Author:

The authors present a HIV immunogen design based on attaching SOSIP trimers to a two component, self-assembling NP system. The NP design used in this manuscript was developed by Neil King's group (Bale et al., Science 2016). The authors show that the SOSIP trimer-NPs can be easily assembled and purified while maintaining the biophysical properties of parental SOSIP trimer. The purified NPs are homogenous, stable, and the authors solved a cryo-EM structure to confirm the proper assembly of NP with 20 SOSIP trimers being presented on the surface.

After that, the authors select BG505 and ConM SOSIP-NPs for binding and immunogenicity experiments. The authors use SPR to show that the antigenicity of SOSIP trimers presented on NP surface is partially maintained. bNAb's targeting the apex-proximal epitopes can still bind to NP-presented SOSIP trimers like they bind to parental SOSIP trimers. However, the accessibility of bNAb epitopes near the trimer base has decreased compared with parental SOSIP trimers, probably due to steric hindrance.

The authors then test the immunogenicity of SOSIP trimer-NP in comparison with parental SOSIP trimers and SOSIP trimer-ferritin. Both BG505 and ConM SOSIP trimer-NPs show increased B cell activation. But for animal experiments, only ConM SOSIP-NP shows significantly improved immunogenicity. The BG505 SOSIP-NP induces even lower binding titers in comparison to parental BG505 SOSIP. The authors attribute this to restricted accessibility of the immunodominant base-proximal epitopes on BG505 SOSIP and use competition ELISA to prove that. Besides, epitope mapping suggests ConM SOSIP-NP induced responses are focused on trimer apex. Overall, these results suggest only SOSIP trimers with apex-proximal immunodominant epitopes benefit from NP presentation.

In summary, the I53-50NP system used in this manuscript shows advantages and can be used as a good platform for immunogen design in general. However, the NP system also generates steric

hindrance such that the accessibility to base-proximal epitopes on presented antigens will be limited. Therefore selecting antigens with exposed immunodominant epitopes is critical while using this type of platform. The manuscript addresses important points for people in immunogen design and HIV biology fields. However, the following issues should be discussed or fixed before the manuscript being accepted for publication.

1. Figure 1b, 2a. The authors should indicate which protein or complex each band represents like in Supplementary Fig 2e.

2. Supplementary figure 1 only show micrographs of negative stain EM. Should also have a panel showing the cryo EM micrographs.

3. Supplementary figure 3b, the BN-PAGE gel actually didn't show much. Other results are good enough to indicate that the BG505 SOSIP-I53-50NP can withstand a freeze-thaw cycle at -80 degree. I suggest remove Supplementary figure 3b.

4. Due to the flexibility of the linker, the outside SOSIP trimer part is poorly resolved compared with the NP part, which is typical for this type of cryo EM structures. But I suggest that the author should include a local resolution figure in Supplementary figure 1. Also I hope the authors could provide the cryo-EM map together with the fitted coordinates to me and other reviewers in the rebuttal. If the manuscript were approved, the authors should upload the cryo-EM map together with the fitted coordinates to EMDB and PDB.

5. Figure 2c. It seems the left panel shows sharpened map of the NP core part and the right panel shows low pass filtered map of the whole SOSIP-NP. I suggest use the sharpened map for both but a lower contour level for the right panel to reveal the SOSIP trimer parts. Also I suggest show these two maps side by side in Chimera such that they are at the same scale.

6. Main text line 153, should mention that the authors were only able to refine the atomic model of the NP core part, but not the SOSIP trimer part.

7. Main text line 178, cryoprotectant is a typo, should be cryoprotectant. Also from Supplementary Fig 3e I cannot see the difference between freeze-thaw in sucrose and freeze-thaw in cryoprotectant. Just from the negative stain EM micrographs I didn't see why AMC011 SOSIP-I53-50NP requires cryoprotectant to maintain its integrity. The authors should elaborate on that or remove this sentence.

8. Supplementary Figure 4a and 4b show binding of IgG to immobilized trimer/trimer-NP. The results basically confirmed strong binding by bNAbs and weak binding by non-NAbs. Then the authors show binding of Fab to immobilized trimer/trimer-NP. By using Fabs instead of IgGs the authors can avoid the avidity effect from IgG and get more quantitative results from SPR. However, the authors spend a lot of effort to explain the lower degree of binding by IgG to NP in the figure legends of Supplementary Figure 4. I don't think it's necessary since the overall purpose of these experiments is just to qualitatively demonstrate that the NP presented trimers have some native antigenicity. I suggest shorten those explanations in Supplementary Figure 4 legends and just say that this could be due to the bivalent binding of IgG and/or the reduced accessibility of epitopes near the base of NP-bound trimer.

9. Similarly, in Figure 4a and 4b the authors showed binding of trimer/trimer-NP to immobilized IgG. In this setup the mass-transport limitation and the avidity effect of IgG will affect the simple Langmuir modeling of SPR sensorgrams. The authors address this in supplementary information and I don't

think this will affect the conclusion in Figure 4b and main text line 218-220. However, I do suggest remove the Langmuir-model fit black lines in Figure 4a and in Figure 4 legend.

10. Main text line 286, the sentence is confusing and sounds like the authors injected I53-NP core itself to see the immune response. Supplementary Figure 6b actually tested whether trimer-NP can elicit Abs against the I53-50NP core. The authors should rewrite this sentence.

Reviewer #1

The manuscript addresses an important point in vaccine-design: the development of nanoparticles that present well-ordered multimeric recombinant protein immunogens. The design and experimental aspects of the project are exceptional. The successful production of self-assembling nanoparticles expressing a complex and notoriously unstable HIV envelope protein, is very impressive. The immunization studies are well controlled. I have no concerns with the interpretation of the results or with the conclusions made by the authors.

We thank the reviewer for these kind remarks on our manuscript.

Although the study is solid, the significance of its main immunological finding (enhanced development of autologous nAbs) is questionable. The goal of HIV envelope-based immunizations is to elicit tier 2, heterologous nAbs. Such nAbs were not developed in the present study, despite the differential exposure of epitopes at the base of SOSIP molecules expressed on the nanoparticles and the altered immunogenicity of that protein when expressed on nanoparticles versus in a soluble form. Since the first immunogenicity study with soluble BG505 SOSIP, many groups have reported that by optimizing the design of the immunogen, or by multimerizing it, an improvement in the development of autologous nAbs is achieved. Yet, heterologous nAbs have not been generated, as is the case here. It is therefore uncertain whether such immunogen-optimization efforts are necessary for the development of tier 2, heterologous nAbs by SOSIP immunogens. It is also questionable if the development of autologous nAbs by SOSIP immunogens will lead (or is required for) to the development of heterologous nAbs. In fact, in the only case where heterologous nAbs were elicited by BG505 SOSIP (Nature 2017) the kinetics of heterologous nAbs were similar to those of the autologous nAbs.

Indeed, although several studies, including our own, have yielded inconsistent and low levels of heterologous NABs, nobody has yet induced heterologous Tier 2 NABs at appreciable titers and consistencies. Achieving that would be a major breakthrough in HIV-1 vaccine research and not one we made here. However, we are convinced that particulate presentation of Env trimers will be part of the answer to solving the HIV-1 bNAb problem, and report considerable progress in this area in the current manuscript. A key aspect of the work is our demonstration that the detailed geometry of epitope presentation on the icosahedral nanoparticle platform has a significant effect on the neutralizing antibody titers obtained upon immunization. Our detailed examination of this phenomenon provides, to our knowledge, the first convincing case in the literature and we believe this will be an important guidepost for nanoparticle immunogen design efforts moving forward. Although it is not unreasonable to think that perhaps this principle will be most important for viruses like HIV-1 since the Env trimer has evolved such sophisticated mechanisms to evade robust neutralizing antibody responses, it could also be important in other efforts where focusing responses on a critical conserved epitope (e.g., the RBS or stem of influenza hemagglutinin). Finally, the manuscript also clearly demonstrates the advantages of *in vitro* assembly of the two-component nanoparticle system for presenting a homogeneous array of native-like Env trimers. This is an advance over commonly used homomeric nanoparticles such as ferritin.

Fig 4c. Why was there no VRC01 B cell activation by SOSIP trimers?

We thank the reviewer for pointing out this apparent inconsistency as SOSIP trimers bind strongly to VRC01 in ELISAs and SPR experiments. SOSIP trimers did also activate VRC01 B cells, but only when used at higher concentrations than the concentration used in the experiment shown, except when the trimers are presented on nanoparticles. Similarly, in the first study that describes this assay, no B cell activation was observed with 2G12 B cell lines at 5 µg/ml of Env trimer (Ota et al. 2012, J Immunol.). This suggests that, although binding can be observed by SPR experiments (where the avidity of immobilized IgGs likely play a large role), it does not necessarily mean that the Env trimer can crosslink and surpass the activation threshold of the B cell receptor.

Fig 6. It is unclear whether the presentation of SOSIP on the I53-50NP rather than on ferritin particles alters the epitope specificity of the autologous neutralizing antibody responses. In other words, is there an advantage of using I53-50NP particles?

We agree with the reviewer that the data in Fig. 6 show that we do not observe significant differences in apical epitope specificity between I53-50 and ferritin. However, we do not feel that this figure provides the key comparison between these two nanoparticle scaffolds. Rather, we have presented data for superiority of the I53-50NPs in Fig. 4c (stronger activation of three bNAb expressing B cell lines); Fig 5b (stronger induction of ConM binding Ab titers over time); Fig. 5c&d (stronger induction of ConM neutralization titers over time); Fig. 5e (weaker induction of undesirable SF162, MW965, MN.3 neutralization) and conclude that I53-50NPs provide a superior vaccine platform than ferritin NPs. We discuss this in lines 395-409 of the discussion section, where we compare and contrast our data with previous studies using ferritin NPs.

Line 307 is incomplete

We could not identify which words were missing, but did rephrase the sentence to enhance clarity (now lines 314-317).

Lines 325 and 328. Supplemental figure 6d does not present data discussed in this paragraph.

We thank the reviewer for pointing out that this is unclear. We have clarified which plots (which are now in Supplementary Fig. 7) these lines refer to by adding (Supplementary Fig. 7b, left) to line 346 and (Supplementary Fig. 7b, right) to line 349.

Reviewer #2

Brouwer et al. reported a combination of SOSIP trimer and self-assembled protein nanoparticles to increase the B cell activation and to shape the immunogenicity towards HIV-1 Env apex-proximal epitopes while suppressing the non-NAb responses. In my opinion, the advantages and disadvantages of using SOSIP-nanoparticles as immunogens were made clear through this study: nanoparticles could withstand freeze-thaw cycle, native-like antigenicity was extensively characterized with bNAbs and SPR, various genotypes of SOSIP were tested, etc. are all merits of this study; but nanoparticles also introduced steric clashes (hence the limited accessibility to epitopes below CD4bs), most neutralization effects were shown towards ConM and ConM is only Tier-1 virus, and the nanoparticle core proteins elicited unnecessary immune responses. I must say the paper started as an insightful and

exciting study but ended with a bit disappointment (because we cannot say the responses against Tier-2 ConS virus was improved when compared to that of SOSIP alone). Overall, I support this publication, but I would like to raise several points that might require further input from the authors to enhance the potential impact of this manuscript:

We thank the reviewer for the positive feedback.

Protein design rationale SOSIP-I53-50 is composed of three parts: Env trimer, NP I53-50, and the linker between them. Since the design of NP I53-50 and the stabilization of SOSIP trimer have been elaborated elsewhere, it may be worth explaining more about how to choose the linker. Specifically, the author said the docking protocol minimized the distance between SOSIP and NP, but then chose a seamlessly arbitrary linker: GGS GGSGGSGGSEKAAKAEAAARK. What is the optimized linker length and how the current linker fit into that?

The linker we used comprises two sections: a flexible N-terminal section (GGSGGSGGSGGS) and a helical C-terminal section (EKAAKAEAAARK). The C-terminal section serves the purpose of extending the N-terminal helix of the I53-50A subunit "up" to the exterior surface of the nanoparticle, as naturally it only extends about halfway "up" the trimer parallel to the three-fold symmetry axis. The flexible N-terminal section of the linker then serves to cover the distance between the termini of the docked Env and I53-50A trimers (including the I53-50A helical extension). In our docked model, this distance is 16 Å. For flexible linker design, we often use a heuristic of 1-2 amino acid residues per 2 Å linear distance, hence the 12-residue flexible linker we used here. We have now explained this in the text (lines 474-482).

The flexibility of the linker seems to be the reason that SOSIP was poorly resolved in cryo-EM. Is there any rationale for preference of a flexible linker than a rigid one?

Generally, when two proteins are fused genetically, flexible linkers are used to allow independent folding of the two domains. We have now explained this in the text (lines 477-480). Whether or not a rigid connection between the nanoparticle scaffold and the displayed antigen would affect immunogenicity is an interesting question that is beyond the scope of the current study, but one that we look forward to studying in future work.

The reasons for using I53-50 as the scaffold seems vague (l. 100-102). For example, I could say I32-19, -06, -28 (among many designs previously reported by the author) might better resemble the diameter of HIV virion (~120 nm), and they all contain larger hollow interior that can be functionalized than then I53-50 does.

We thank the reviewer for raising this issue, which we did not address adequately in the original manuscript. We screened several of our designed nanoparticle components for expression when fused to BG505 SOSIP and found that the trimeric component of I53-50 (I53-50A) secreted at higher levels than any other scaffold. We also knew from biophysical characterization of several of our nanoparticles that the I53-50A trimer is hyperstable, and reasoned that this might help with proper folding and secretion of the fused Env antigen. Furthermore, in a parallel effort, we had been successful fusing the prefusion RSV F antigen DS-Cav1 to I53-50A (Marcandalli et al., *Cell* 2019), further supporting continued

investigation of this construct. We have now included an explanation why we choose the I53-50 platform (lines 97-102).

The Env spikes on the surface of HIV are sparsely distributed, unlike the dense packing on NP as revealed by the EM structure. I am not sure whether the denser alignment of Env would be good or bad for its immunogenicity, maybe the authors can provide some insights.

It is generally accepted that the sparse density of Env on the surface of HIV is in fact one cause of its poor immunogenicity during HIV infection. The probable mechanistic explanation is that the sparse Env density precludes B cell receptor cross-linking (Schiller & Chackerian 2014, *PLoS Path.*; Klein & Björkman 2010, *Plos Path*). The corollary is that one would not wish to emulate this in a vaccine. In contrast, higher antigen density is desired for efficient B cell receptor cross-linking and B cell activation. We have now alluded to this in the introduction and included the Schiller & Chackerian and Klein & Bjorkman references (lines 86-87).

L. 107: why were there four variants of I53-50A? What mutations were introduced? Were these mutations of I53-50 also used for ConM, AMC011, ZM197M constructs to increase the overall stability of NPs?

The three additional variants of the prototype trimeric I53-50A component (A.1NT1, A.1NT2 and A.1PT1) were designed as part of a previous effort to drive encapsulation of biomolecules in the I53-50 interior (Bale et al., *Science* 2016). The sequences of these three variants are shown in the Supplement and the mutations are highlighted in green for clarity. While screening genetic fusions to BG505 SOSIP, we found that the A.1NT1 variant showed the highest expression of cleaved fusion protein (lines 106-113). However, such an effect was not observed with ConM, ZM197M and AMC011 trimers. Therefore, we used the prototypic I53-50A for the latter three trimers and I53-50A.1NT1 in combination with BG505 trimers. We regret the inconsistent use of nomenclature and have modified the text in various places for clarity.

Why was ZM197M-NP construct discarded at later stages? Based on Fig. 3 and Fig. S3, ZM197M-NP assembled equally well and were stable enough. Similar strong binding by bNAbs to ZM197M was also observed in Fig. S4.

Indeed the ZM197M nanoparticles assembled very efficiently and were thermostable. We have certainly not discarded them as vaccine candidates/components, nor have we discarded the AMC011 nanoparticles. However, for generating proof-of-concept *in vivo* data, we chose to work on the BG505 and ConM genotypes because 1) the availability of abundant historic data to provide a frame of reference, 2) the known location of the immunodominant epitopes, and 3) the different locations of the immunodominant epitopes. The immunodominant epitopes for ZM197M and AMC011 are currently unknown. We have now made this clear in the results section lines 245-251.

Comparing Fig. 4 to Fig. S4 could lead to some confusion: in Fig. 4 VRC01 or PGT122 bNAbs bound NP with higher capacity but in Fig. S4 VRC01 or PGT122 exhibited stronger interactions towards the parental trimer. I do not know whether it is because IgGs were immobilized in Fig. 4 while NPs were immobilized in Fig. S4, but I think the readers may feel confused as well. Table 2 was appreciated, but I would suggest the authors also make a dot

plot of the simulated KD to clearly show which is higher and which is lower, for easy comparison.

The higher signal obtained from nanoparticles compared to trimers against immobilized VRC01 and PGT122 does not represent stronger or more binding, but results from the nanoparticles' greater mass. The signal is proportional to the mass bound and for this reason we included Fig. 4b. We understand that this may have been unclear and have therefore altered the text to clarify (lines 211-219).

We thank the reviewer for the suggestion. However we believe a dot plot would be less direct and it would require some kind of quite complicated color code. In the right-most column of Table 2 the values to be compared are given in neighboring cells, the one underneath the other. In addition, by putting all the values in a table we also emphasize that Kd is not necessarily more interesting than the other measurements. Finally, none of these values is simulated: all are fitted to empirical data. The visual impression of the similarity among all binding parameters, fitted or calculated in the respective NP-trimer comparisons, can be gleaned from Fig. S4c. This is explained in the text (lines 200-204).

Another confusion might happen for the last four paragraphs of Results. To map the immunodominant epitope of BG505-NP and ConM-NP, respectively, the authors applied different assays. For BG505, the knock-in virus was used for neutralization; for ConM, a competition neutralization assay was designed by incubating sera with SOSIP while the virus was not changed. In competition ELISA, different apex-specific bNAbs were used (PGT128 and VRC01 for BG505 PG16, and gl-PG9 for ConM). It seems there is a lack of consistency for characterizing the two NP constructs. Maybe I am missing some information or background knowledge, but the readers may feel similarly misled too, for example, could the pre-incubating sera with BG505 SOSIP be applied to BG505 neutralization?

This is a very good point and we regret the impression of misleading the reader. The reason for some of the differences in the analytical tools used is that the BG505 and ConM/ConS viruses and the two corresponding trimers behave differently. For example, they have different immunodominant epitopes and that is one reason that we selected both of them. Nevertheless, we did try to harmonize the methodologies used for both immunogens and viruses. Accordingly, we used virus mutants to map the responses induced by both BG505 and ConM immunogens (BG505 virus mutants for BG505 immunogens, ConS mutants for ConM immunogens; and diagnostic mutants were selected based on the expected immunodominant responses; Klasse *et al.* 2018, *PLoS Path.*, Bontjer *et al.* 2019, *Nat. Comm*). Similarly, we used competition ELISAs for both BG505 and ConM immunogens, but the analyte antibodies were chosen based on the described (immunodominant) epitopes, and were therefore different. However, we have added VRC01 and PGT128 to the competition ELISA of ConM recipients so that it is more consistent with the apex-specific Abs used for BG505 (see Fig. 6e).

We agree with the reviewer that the serum pre-incubation experiments provide robust data for neutralization specificity and in the previous version of the manuscript we only included such data for ConM, not BG505. We have now generated and included the requested data for BG505. Accordingly, we performed BG505 neutralization depletion experiments with BG505 SOSIP trimers versus BG505 SOSIP trimers in which the immunodominant N241/N289 glycan hole was filled in. All trimers had the D368R mutation to prevent neutralization by the depleting trimer. The results corroborated the finding that, in contrast to BG505 SOSIP trimer recipients, the 241/289 glycan hole is not the immunodominant epitope

for BG505 SOSIP-I53-50NP recipients. These data are now included in Supplementary Fig. 7a. and described in the accompanying text (lines 327-339).

Although the authors suggested that self-assembling proteins are better than VLPs or liposomes, the adjuvant used in the immunization was a liposome. I wonder whether the hydrophobic interactions in the liposome would interfere with NPs. I can not see much from Fig. S6a about structural integrity as claimed, I can only see the morphology of particles is different from that of Fig. S3c.

The reviewer wonders whether the hydrophobic interactions in the liposome would interfere with the NPs. We saw no evidence of such interference. The GLA-LSQ liposomes were formulated separately from the nanoparticles and the two components were mixed prior to use. The liposomes would need to fall apart before there would be free lipids to form hydrophobic interactions with the nanoparticles. The EM images in Supplementary Fig. 6a do not show liposomes that are falling apart nor do they show micelles that would form if the liposomes did fall apart. Furthermore, no traces of unassembled components could be observed by NS-EM indicating that the nanoparticles maintain their integrity in the presence of GLA-SQ.

Recent studies using trimer-conjugated liposomes (e.g. Tokatlian et al., 2019) also suggested this liposome formulation can trigger greater antibody responses and enhance humoral immunity. So I am not sure protein NP is better than liposome.

While our study included a head-to-head comparison with ferritin nanoparticles, we did not include such a head-to-head comparison with liposomes. We have therefore removed claims of superiority over liposomes from the manuscript.

It seems L. 59 directly copy the third sentence in the abstract; I suggest the authors could introduce the difference between NP and VLP at this place for readers who are not familiar with different formulas.

We have now modified the sentence as follows: “Displaying antigens in a particulate array on synthetic NPs, or alternatively, on virus-like particles.....”

Fig. 1b said ‘I53-50A1. NT1’, the main text said ‘I53-50A. INT1’

We thank the reviewer for pointing out this inconsistency and have corrected it.

Fig. S3a, DSF plot, it seems the I53-50NP alone (black) did not have a melting peak at 82 degrees, or the melting temperature is larger than 100 degrees. If the latter case is true, does it mean fusion with SOSIP reduced the stability of NPs?

I53-50 is in fact a remarkably stable nanoparticle, and we have not been able to accurately determine a melting temperature because it only starts to unfold at >90 °C according to several measures, including circular dichroism (not shown) and nanoDSF (Supplementary Fig. 3a). The minimum in the plots in Supplementary Fig. 3a near 80-85 °C does appear to represent a change in the fluorescence of the nanoparticle scaffold, suggesting that fusion of SOSIP trimers does slightly destabilize the I53-50NP. This is now discussed in the text (lines 142-144).

Fig. 3b legend said Protein Stabilizing Cocktail, while Fig. S3e said sucrose or PSC, so which one is the cryoprotectant?

Both sucrose and PSC are cryoprotectants. We have modified the text (line 183) to make this more clear. Supplementary Fig. 3d now contains three panels: AMC011 nanoparticles without a cryoprotectant, with sucrose or with PSC. We chose to show AMC011 SOSIP-I53-50NPs in the presence of PSC for the main figure.

L. 183, it is better to state where the His-tag located.

We added descriptions to the sentence (now line 189-190).

Reviewer #3

In summary, the I53-50NP system used in this manuscript shows advantages and can be used as a good platform for immunogen design in general. However, the NP system also generates steric hindrance such that the accessibility to base-proximal epitopes on presented antigens will be limited. Therefore selecting antigens with exposed immunodominant epitopes is critical while using this type of platform. The manuscript addresses important points for people in immunogen design and HIV biology fields. However, the following issues should be discussed or fixed before the manuscript being accepted for publication.

We thank the reviewer for the positive and accurate summary of our manuscript.

1. Figure 1b, 2a. The authors should indicate which protein or complex each band represents like in Supplementary Fig 2e.

We have added the requested labels to Fig. 1b and 2a.

2. Supplementary figure 1 only show micrographs of negative stain EM. Should also have a panel showing the cryo EM micrographs.

The requested panel is now included as Supplementary Fig. 1c.

3. Supplementary figure 3b, the BN-PAGE gel actually didn't show much. Other results are good enough to indicate that the BG505 SOSIP-I53-50NP can withstand a freeze-thaw cycle at -80 degree. I suggest remove Supplementary figure 3b.

We agree and have removed Supplementary Fig. 3b.

4. Due to the flexibility of the linker, the outside SOSIP trimer part is poorly resolved compared with the NP part, which is typical for this type of cryo EM structures. But I suggest that the authors should include a local resolution figure in Supplementary figure 1. Also I hope the authors could provide the cryo-EM map together with the fitted coordinates to me and other reviewers in the rebuttal. If the manuscript were approved, the authors should upload the cryo-EM map together with the fitted coordinates to EMDB and PDB.

We thank the reviewer for their suggestions. We have added the local resolution figure in Supplemental figure 1 and uploaded the cryo-EM map together with the fitted coordinates to

EMDB (EMD-20261) and PDB (6P6F).

5. *Figure 2c. It seems the left panel shows sharpened map of the NP core part and the right panel shows low pass filtered map of the whole SOSIP-NP. I suggest use the sharpened map for both but a lower contour level for the right panel to reveal the SOSIP trimer parts. Also I suggest show these two maps side by side in Chimera such that they are at the same scale.*

We have adjusted the two maps to be on the same scale however we think it would be best to keep the lowpass filtered map displaying the trimer density. First, the trimer density in the sharpened map is so diffuse that it is difficult to interpret at low thresholds and makes for an aesthetically unpleasing figure. It also obscures the view of the underlying nanoparticle core. Also, since we used Segger (Pintilie, J. Struct. Biol. 2010) to segment and color the map components, using a sharpened map would prohibit accurate segmentation of the trimers into distinct components.

6. *Main test line 153, should mention that the authors were only able to refine the atomic model of the NP core part, but not the SOSIP trimer part.*

We have modified the sentence (now line 157) accordingly.

7. *Main test line 178, cryoprotectant is a typo, should be cryoprotectant. Also from Supplementary Fig 3e I cannot see the difference between freeze-thaw in sucrose and freeze-thaw in cryoprotectant. Just from the negative stain EM micrographs I didn't see why AMC011 SOSIP-I53-50NP requires cryoprotectant to maintain its integrity. The authors should elaborate on that or remove this sentence.*

We thank the reviewer for catching this typo. We understand that this part may not have been clear to the reviewer. Both sucrose and PSC were used as cryoprotectants. We have clarified now this in line 183. We have also added a micrograph in Supplemental Fig. 3d that shows AMC011 SOSIP-I53-50NP after freeze-thawing in the absence of a cryoprotectant.

8. *Supplementary Figure 4a and 4b show binding of IgG to immobilized trimer/trimer-NP. The results basically confirmed strong binding by bNAbs and weak binding by non-NAbs. Then the authors show binding of Fab to immobilized trimer/trimer-NP. By using Fabs instead of IgGs the authors can avoid the avidity effect from IgG and get more quantitative results from SPR.*

However, the authors spend a lot of effort to explain the lower degree of binding by IgG to NP in the figure legends of Supplementary Figure 4. I don't think it's necessary since the overall purpose of these experiments is just to qualitatively demonstrate that the NP presented trimers have some native antigenicity. I suggest shorten those explanations in Supplementary Figure 4 legends and just say that this could be due to the bivalent binding of IgG and/or the reduced accessibility of epitopes near the base of NP-bound trimer.

We can relate to the point that the legend of Supplementary Fig. 4 was too detailed. The purpose of our SPR studies was not only to show “some native antigenicity” but to describe the similarities and differences with the greatest possible precision both qualitatively and quantitatively. We used only Supplementary space for presenting these findings, but the passage from “The lower degree of binding by IgG to NPs could be due...” to the end of the legend has now been clarified and cut by 40%. We believe the findings justify mention and

explanation as 1) they pertain to the interaction of NPs with individual B cell receptors compared with the crosslinking of multiple B cell receptors; and 2) taken together they give the strongest support possible for antigenic preservation of the trimers in the NP context.

9. Similarly, in Figure 4a and 4b the authors showed binding of trimer/trimer-NP to immobilized IgG. In this setup the mass-transport limitation and the avidity effect of IgG will affect the simple Langmuir modeling of SPR sensorgrams. The authors address this in supplementary information and I don't think this will affect the conclusion in Figure 4b and main text line 218-220. However, I do suggest remove the Langmuir-model fit black lines in Figure 4a and in Figure 4 legend.

We agree with the referee that because of the mass transport-limitation – more marked for NPs than trimers, which is a reasonable confirmatory finding – the fitted Langmuir function should not appear in the figure without a caveat. We have now noted this in the legend of Fig. 4a. If we had removed the curves, however, then what we refer to in this paragraph and describe in the more extended mass-transport-limitation section in supplement (lines 139-149) would not be shown anywhere. To maximize transparency we have therefore instead added cross-references in the legend of Figure 4a and Supplementary Methods (lines 141-143).

10. Main text line 286, the sentence is confusing and sounds like the authors injected I53-NP core itself to see the immune response. Supplementary Figure 6b actually tested whether trimer-NP can elicit Abs against the I53-50NP core. The authors should rewrite this sentence.

We have rewritten the sentence as follows: “To test whether the I53-50 core of the SOSIP-I53-50NPs was immunogenic, we measured Ab binding titers against I53-50NPs devoid of SOSIP trimers.”

Reviewers' Comments:

Reviewer #2:

Remarks to the Author:

I believe my concerns have been addressed and the manuscript was significantly improved. I also appreciate the author's elaboration of protein design and adjuvant formulation rationales in the revised manuscript.

Since the last review, there were advancements in the fields:

Zhu et al., Nature Communications, 10:948 2019 (very similar story);

Escolano A et al., Nature 2019;

Cirelli KM et al., Cell 2019.

It is important that the author could update the background parts with these references.

Reviewer #3:

Remarks to the Author:

The authors have adequately responded to my comments and I believe the paper is now acceptable for publication. I appreciate that the authors uploaded the EM map and fitted coordinates to EMDB and PDB.

Haoqing Wang

REVIEWERS' COMMENTS:

Reviewer #2 (Remarks to the Author):

I believe my concerns have been addressed and the manuscript was significantly improved. I also appreciate the author's elaboration of protein design and adjuvant formulation rationales in the revised manuscript.

Since the last review, there were advancements in the fields:

Zhu et al., Nature Communications, 10:948 2019 (very similar story);

Escolano A et al., Nature 2019;

Cirelli KM et al., Cell 2019.

It is important that the author could update the background parts with these references.

We thank the reviewer for the feedback. Indeed some interesting new advancements have been made since the last review. We have now incorporated the references of Cirelli et al. and Escolano et al. in our manuscript. The first, stresses the need for further research in delivery systems as they show how a slow-delivery system can significantly increase the immunogenicity of Env trimers; the latter, describes Env trimers that can be in vitro assembled on VLPs using the previously described SpyTag-SpyCatcher system. Although also an advancement to the field, we feel referencing to the paper by Zhu et al. would be less appropriate. The Zhu paper mentions a nanoparticle that presents a small domain of HIV-1 Env. However, our paper is very strictly focused to nanoparticle immunogens for native-like full-length structures of Env. For the same reason we have not cited other high-impact papers that describe nanoparticles presenting domains or peptides of Env such as Jardine et al. 2013 that describes eOD-trimers on lumazine synthase nanoparticles.

Reviewer #3 (Remarks to the Author):

The authors have adequately responded to my comments and I believe the paper is now acceptable for publication. I appreciate that the authors uploaded the EM map and fitted coordinates to EMDB and PDB.

Haoqing Wang

We thank the reviewer for the feedback.